JCB | Journal of Cell Biology

# Giantin is required for intracellular N-terminal processing of type I procollagen

Nicola L. Stevenson[1], Dylan J.M. Bergen[2,3,4], Yinhui Lu[5,6], M. Esther Prada-Sanchez[1,2], Karl E. Kadler[5,6], Chrissy L. Hammond[2], and David J. Stephens[1]

Knockout of the golgin giantin leads to skeletal and craniofacial defects driven by poorly studied changes in glycosylation and extracellular matrix deposition. Here, we sought to determine how giantin impacts the production of healthy bone tissue by focusing on the main protein component of the osteoid, type I collagen. Giantin mutant zebrafish accumulate multiple spontaneous fractures in their caudal fin, suggesting their bones may be more brittle. Inducing new experimental fractures revealed defects in the mineralization of newly deposited collagen as well as diminished procollagen reporter expression in mutant fish. Analysis of a human giantin knockout cell line expressing a GFP-tagged procollagen showed that procollagen trafficking is independent of giantin. However, our data show that intracellular N-propeptide processing of pro-α1(I) is defective in the absence of giantin. These data demonstrate a conserved role for giantin in collagen biosynthesis and extracellular matrix assembly. Our work also provides evidence of a giantin-dependent pathway for intracellular procollagen processing.

## Introduction

The golgins are a family of coiled-coil domain proteins that extend out from the surface of the Golgi apparatus to tether transport vesicles and other Golgi membranes (Munro, 2011). The largest member of this family, giantin, is a tail-anchored membrane protein with a predicted 37 cytosolic coiled-coil domains (Linstedt and Hauri, 1993; Seelig et al., 1994). These structural features are key attributes for a membrane tether; however, to date, no tethering function for giantin has been identified. Indeed, giantin loss does not block anterograde transport (Lan et al., 2016; Stevenson et al., 2017) and may in fact accelerate it (Koreishi et al., 2013). Most studies also agree that giantin is not essential to maintain Golgi morphology (Koreishi et al., 2013; Lan et al., 2016; Puthenveedu and Linstedt, 2001; Stevenson et al., 2017), although it may inhibit lateral tethering between cisternae (Satoh et al., 2019; Stevenson et al., 2017). Discrepancies between these studies are likely due to variation in levels of depletion (Bergen et al., 2017), in genetic compensation (Stevenson et al., 2017), and/or functional redundancy with other golgins (Wong and Munro, 2014).

The most consistent observation from published work is that giantin is required to regulate glycosylation (Kikukawa et al., 1990; Koreishi et al., 2013; Lan et al., 2016; Petrosyan et al., 2014; Stevenson et al., 2017) and ECM formation (Katayama et al., 2018; Kikukawa and Suzuki, 1992; Lan et al., 2016). Highly selective defects in O-glycosylation have been reported following knockout (KO) of the GOLGB1 gene encoding giantin in cells (Stevenson et al., 2017), zebrafish (Stevenson et al., 2017), and mice (Lan et al., 2016). Enzyme distribution (Petrosyan et al., 2014) and surface glycosylation patterns (Koreishi et al., 2013) are more generally affected following siRNA depletion. The secretion of ECM proteoglycans and collagen can also be affected (Katayama et al., 2018; Kikukawa et al., 1990). The primary phenotype shared by all GOLGB1 KO animal models is the abnormal development of craniofacial structures, while species-specific phenotypes include short limbs in rats (Katayama et al., 2011) and ectopic mineralization of soft tissues in zebrafish (Stevenson et al., 2017). Giantin is therefore important for skeletal development, and defects in ECM structure likely underlie all these phenotypes.

In light of these observations, we hypothesized that giantin may regulate secretion of the primary protein component of skeletal ECM, fibrillar type I collagen. In mammals, this is predominantly built from heterotrimeric molecules composed of two pro-α1(I) chains (encoded by the COL1A1 gene) and one pro-α2(I)

[1]Cell Biology Laboratories, School of Biochemistry, Faculty of Life Sciences, University of Bristol, Bristol, UK; [2]School of Physiology, Pharmacology and Neuroscience, Faculty of Life Sciences, University of Bristol, Bristol, UK; [3]Musculoskeletal Research Unit, Translational Health Sciences, University of Bristol, Bristol, UK; [4]Bristol Medical School, Faculty of Health Sciences, University of Bristol, Southmead Hospital, Bristol, UK; [5]Wellcome Centre for Cell-Matrix Research, Faculty of Biology, Medicine and Health, University of Manchester, Manchester, UK; [6]Manchester Academic Health Science Centre, Manchester, UK.

Correspondence to Nicola L. Stevenson: nicola.stevenson@bristol.ac.uk; David J. Stephens: david.stephens@bristol.ac.uk

chain (encoded by *COL1A2*). Structurally, each chain is made up of a helical domain, composed of Gly–X–Y repeats, flanked by globular N- and C-propeptide domains (Canty and Kadler, 2005). These chains are cotranslationally translocated into the ER lumen, where they are post-translationally modified before folding into a right-handed triple helical molecule. Trimeric procollagen is exported from the ER and transits the Golgi before being secreted from the cell and assembled into fibrils in the ECM (Canty and Kadler, 2005; Canty et al., 2004).

Prior to fibrillogenesis, the N- and C-propeptide domains of procollagen are cleaved to promote correct alignment and polymerization. Removal of the C-propeptide is particularly critical as this induces self-assembly of collagen into fibrils (Hulmes et al., 1989; Kadler et al., 1987; Kadler et al., 1990; Miyahara et al., 1984; Miyahara et al., 1982). Retention of the N-propeptide, on the other hand, does not preclude fibril assembly but can affect fibril morphology (Bornstein et al., 2002; Hulmes et al., 1989; Romanic et al., 1992). C-terminal processing is performed by BMP-1/tolloid-like family metalloproteinases (Kessler et al., 1996) while ADAMTS2, -3, and -14 cleave the N-propeptide (Bekhouche and Colige, 2015; Colige et al., 1997; Colige et al., 2002; Fernandes et al., 2001). Meprin α and meprin β have also been implicated in procollagen processing (Broder et al., 2013).

In this study, by using a combination of fin fracture assays in *golgb1* mutant zebrafish and biochemical assays in giantin KO cells, we demonstrate that giantin function is required to facilitate normal fracture repair and for intracellular N-terminal processing of type I procollagen.

## Results

### Homozygous (HOM) *golgb1* mutant fish have a higher incidence of fracture

To investigate the role of giantin in the deposition of skeletal ECM, we examined our previously published HOM *golgb1^{X3078/-X3078}* mutant zebrafish line for bone defects (Bergen et al., 2017). Focusing on the caudal fin (Bergen et al., 2019), we observed an unusually high number of naturally occurring fractures in the hemirays of HOM individuals compared with WT and heterozygote (HET) siblings. This was seen both in terms of the number of injured fish and the number of fractures per individual. Indeed, at 7 mo old, 76% of HOM fish had acquired at least one fracture compared with just 33% of WT and 27% of HET fish (Fig. 1 A). The mean number of fractures per individual was 0.4 for WT and HETs and 1.8 for mutants. Interestingly, 12% of *golgb1^{X3078/X3078}* mutants had ≥4 fractures. This was never seen in WT or HET siblings. These were often in the proximal half of the fin and occurred either consecutively along one ray or adjacent to each other in parallel rays (Fig. 1 B). There was also one HOM individual carrying fractures that had fused together, indicating aberrant fracture repair with excessive calcification (Fig. 1 B).

### *golgb1* mutant fish show abnormal mineralization patterns during fracture repair

To investigate this further, fractures were experimentally induced in caudal fin hemirays of mutant and WT fish as described

previously (Geurtzen et al., 2014; Tomecka et al., 2019). Injured fish were stained with calcein at different time points to monitor calcification of newly formed bone matrix in the callus. Calcein labeling was first visible in WT fish 4 d post injury (dpi), whereas labeling was already apparent at the fracture site in HOM fish at 2 dpi (Fig. 1, C and D). In both cases, labeling intensity continued to increase over time, peaking at 10 dpi. Calcein fluorescence was substantially greater in the fractures of HOM fish throughout the assay. This is most evident at 10 dpi, when it was ~400% brighter in the mutants. Levels of calcein-accessible calcium are therefore elevated in the fractures of *golgb1^{X3078/X3078}* fish. This shows that premature and enhanced calcification occurs in the mutant fish.

### Expression of the *col1a1* gene is reduced in *golgb1* mutant fractures

Mineralization is dependent on the correct spatio-temporal deposition of a collagen matrix to ensure the right amount of calcification occurs at the right time (Michigami, 2019). We therefore assessed type I collagen expression in the repairing fractures. We crossed *golgb1^{X3078/X3078}* fish with a *col1a1a:GFP* promoter reporter line and performed the same fracture experiments as above but without the calcein stain. Expression of *col1a1a:GFP* at the fracture site relative to healthy bone was highest at 4 dpi in both HET and HOM fractures (Fig. 1, E and F). Expression returned to normal levels by 14 dpi. Interestingly, *col1a1a:GFP* expression was lower in the fractures of HOM mutant fish compared with HETs at both 4 and 7 dpi, when promoter activity was at its peak. This agrees with our previously published RNA sequencing (RNA-seq) data, which show a reduction in *COL1A1* mRNA in *GOLGB1* KO human telomerase-immortalized retinal pigment epithelial (hTERT-RPE1) cells relative to WT (Stevenson et al., 2017). No obvious difference in callous width or Alizarin Red staining (ARS) was observed (Fig. S1, A and B).

### There are no gross changes in collagen structure in the mutant fins

We next looked to see if we could identify a change in collagen synthesis and deposition that would account for the lower expression of *COL1A1* mRNA and decreased GFP signal. The distal third of the caudal fin was amputated and lysed to produce a sample rich in bone collagen to analyze by immunoblotting. Collagen blots were similar between the WT and mutant fins, suggesting collagen is predominantly synthesized and processed normally in the mutants (Fig. S1 C).

We considered the possibility that collagen secretion or structural defects occurring early during synthesis could later be resolved in the tissues as they mature. We therefore decided to test younger, nascent tissue to look for phenotypes. First, we amputated the distal third of the caudal fin and then let it regenerate for 5 d before amputating and lysing the new regenerated tissue (Fig. S1, D and E). Second, we lysed young larvae at 5 days post-fertilization (dpf) to probe collagen early in development (Fig. S1, F and G). In both cases, we failed to see any difference in collagen migration through the gel by immunoblot or by Coomassie stain. The lysates were not decalcified, which

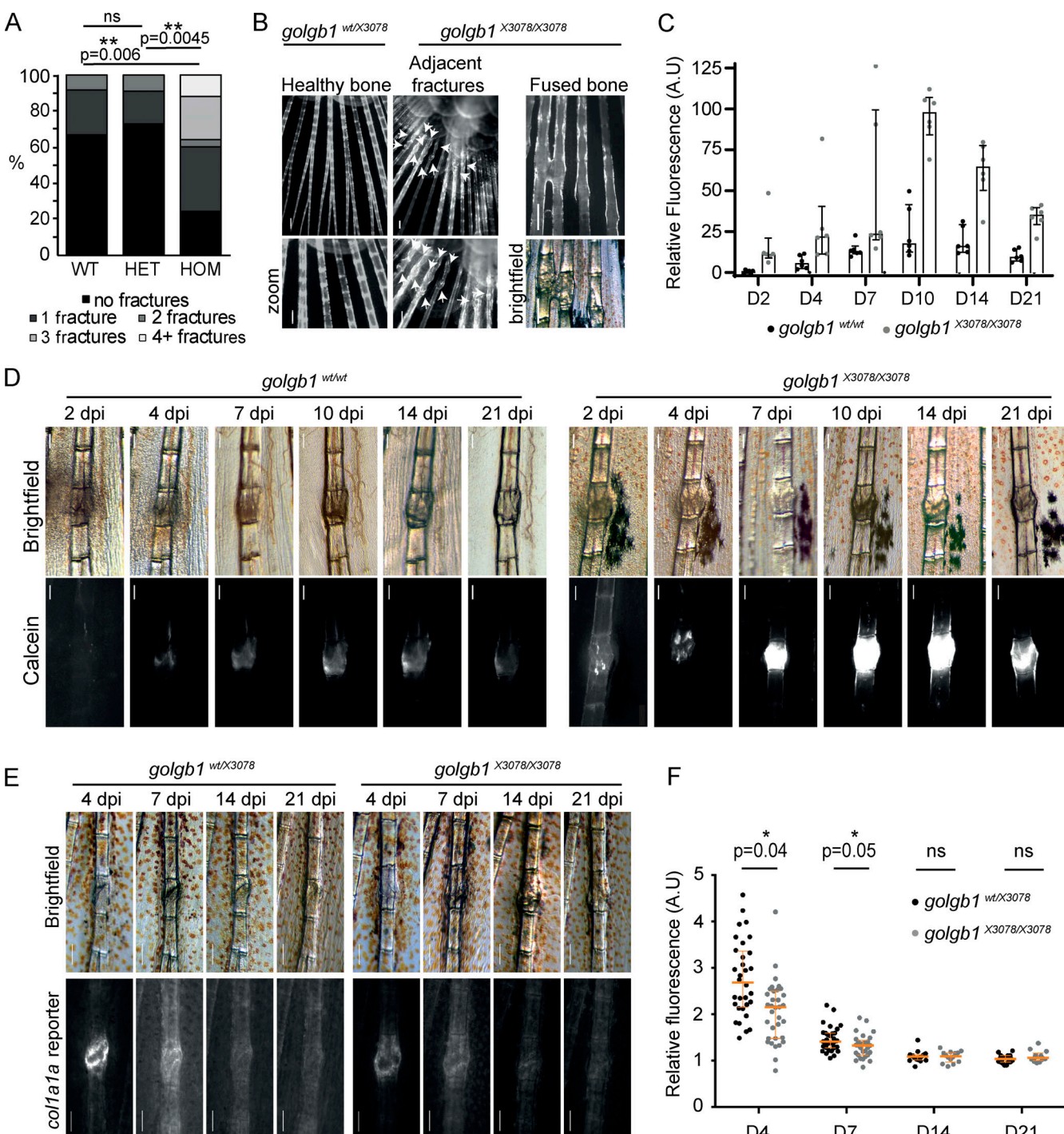

Figure 1. **Fracture defects in *golgb1* mutant zebrafish. (A)** Quantification of the number of fractures naturally found in the caudal fin of WT and *golgb1^{X3078/X3078}* heterozygous and HOM mutant fish at 7 mo old. Data show percentage of fish with × number of fractures (WT = 12 fish, HET = 11 fish, HOM = 25 fish from two independent crosses). Statistics performed with a Mann-Whitney *U* test. **(B)** Fluorescence images of naturally occurring caudal fin defects in *golgb1^{X3078/X3078}* fish stained with ARS. White arrows indicate fractures. Scale bar = 200 µm. **(C and D)** Quantification (C) and representative brightfield and fluorescent images (D) of experimentally induced fractures in *golgb1^{wt/wt}* and *golgb1^{X3078/X3078}* caudal fins on different dpi. Bone was stained with calcein at each time point before imaging. **(C)** Calcein intensity in fractures was measured relative to that of healthy adjacent bones. Lower exposure images than those in D were used for quantification to avoid saturation. Each dot represents one fracture. Bars show median and interquartile range (two fish per line quantified, each with three fractures). **(D)** Proximal end of bone is at the top of image. Scale bar = 200 µm. **(E)** Representative brightfield and fluorescent images of experimentally induced fractures in the caudal fin of *golgb1^{wt/X3078}* and *golgb1^{X3078/X3078}* fish expressing a *col1a1a:GFP* promoter reporter at different time points. Proximal end of bone is at the top of image. Scale bar = 200 µm. **(F)** Quantification of *col1a1a:GFP* signal at the fracture site relative to an adjacent healthy bone. Each dot represents one fracture. Orange bars indicate median and interquartile range. At time points 4 and 7 dpf, 11 fish per line were quantified. At time points 14 and 21 dpf, *n* = 6 HETs and *n* = 4 HOM fish quantified. All data were collected in a single experiment. All P values were calculated with the Mann-Whitney *U* test comparing the means for each fish, three fractures per fish. A.U, arbitrary units.

may explain the absence of an obvious collagen abundance by Coomassie stain.

We next investigated collagen fibril structure in the ECM of mutant bones. Again, on the assumption that we were most likely to see a phenotype in younger tissue, we processed regenerated caudal fins for analysis by transmission EM (TEM). Imaging of tissue from four individuals of each genotype failed to show any consistent differences in fibril structure or abundance (Fig. S1 H); however, there was considerable variation between individuals. Collagen fibrils in the bone had regular contours in cross section, and collagen fibril D-periodicity was normal. The mutants also produced a structurally normal basement membrane in the skin (Fig. S1 I).

### Expression and secretion of type I collagen is elevated in giantin KO cells

Given the absence of gross morphological changes in the maturing collagen matrix of mutant fish, we next focused on the early steps of secretion. For this, we turned to our previously published *GOLGB1* KO hTERT-RPE1 cell line as a more tractable system for studying intracellular events (Stevenson et al., 2017). Our previously published RNA-seq data show that gene expression of *COL1A1* is reduced in giantin KO cells (Stevenson et al., 2017). In contrast, immunoblot analysis shows that KO cells contain higher levels of type I collagen protein relative to WT cells (Fig. 2, A and B). While statistical testing of pooled data does not reveal a detectable difference here, comparison of individual experiments (color coded in Fig. 2 B) shows a consistent increase of pro–α1(I). Interestingly, higher procollagen levels are not due to protein retention since KO cells also secrete similar or higher amounts of type I collagen compared with WT cells, both in absolute terms and relative to the intracellular pool (Fig. 2, C and D). There is also no evidence of collagen overmodification or ER dilation (Stevenson et al., 2017), as would be expected if the collagen had been retained (Ishida et al., 2006). Immunofluorescence labeling of type I collagen in non-permeabilized WT and KO cells verified that this excess of secreted collagen is incorporated into the ECM (Fig. 2, E and F). In fact, the cell-derived matrix from giantin KO cells often contained thicker fibers that were more intensely labeled for type I collagen, but these were difficult to quantify.

### Procollagen trafficking through the early secretory pathway is unaffected by loss of giantin

We next tested whether altered procollagen synthesis and deposition is the result of changes to procollagen trafficking through the secretory pathway. To this end, we generated WT and giantin KO cell lines stably expressing low levels of pro–streptavidin binding protein (SBP)–GFP-COL1A1 (GFP-COL1A1; McCaughey et al., 2019). This construct encodes pro-α1(I) with a GFP and SBP tag inserted upstream of the N-propeptide cleavage site (Fig. 3 A). The SBP tag allows controllable bulk release of the procollagen from the ER using the Retention Using Selective Hooks (RUSH) assay (Boncompain et al., 2012). This assay relies on coexpression of a KDEL-streptavidin hook to bind the SBP and anchor the GFP-COL1A1 in the ER. The addition of biotin releases GFP-COL1A1. The hook is co-expressed from a bicistronic vector with mCherry-sialyltransferase (mCh-ST), which labels the Golgi.

In WT cells, GFP-COL1A1 first accumulates around the mCh-ST–positive Golgi compartment upon release (Fig. 3 B and Video 1). Fixation at this time point followed by immunolabeling of Golgi markers confirmed that this predominantly represents the GFP-COL1A1 entering the cis-Golgi (Fig. 3 E). Shortly after cis-Golgi filling, the GFP-COL1A1 progresses through the Golgi into the mCh-ST–positive cisternae before leaving the Golgi in tubular vesicular carriers (Fig. 3, B and C, arrows). This "short loop" pathway from the ER to cis-Golgi in the absence of discernible carriers has been described previously (McCaughey et al., 2019). Here, we found that GFP-collagen also traverses this pathway in the giantin KO cells (Fig. 3 C and Video 2).

To measure the kinetics of ER-to–cis-Golgi transport, we quantified the time between biotin addition and the appearance of GFP-COL1A1 in the cis-Golgi. Trafficking rates were equivalent in both cell lines (Fig. 3 D), indicating the giantin KO phenotypes are not due to the use of an alternative procollagen trafficking route or faster kinetics.

### Giantin KO cells exhibit defects in processing of type I procollagen

During these experiments, we observed that giantin KO cells stably expressing GFP-COL1A1 were often surrounded by extracellular GFP-positive fibers. Immunofluorescence with a pro-α1(I) antibody identified these as collagen fibrils, indicating that the GFP-tag is incorporated into the ECM (Fig. 4 A). No GFP-COL1A1–positive fibrils could be detected in the matrix of WT cells despite an abundance of collagen, and the expression of GFP alone did not label the ECM.

In our reporter construct, GFP is inserted upstream of the N-propeptide cleavage site and is expected to be removed during procollagen processing (Fig. 3 A). The presence of GFP in the ECM therefore implies that either N-propeptide processing is defective or that free N-propeptides are abnormally interacting with collagen fibrils. To investigate this, we assayed secretion by immunoblotting. In both the media and lysate fractions of WT cultures, a dominant GFP-positive band was discernible at ~60 kD, which corresponds to the size of the tagged N-propeptide (Fig. 4 B). In KO cells, however, the dominant GFP-positive band was detected at ~200 kD. This is consistent with the size of unprocessed, tagged procollagen. No band was discernible at ~60 kD in the KO cells, even after enrichment by immunoprecipitation (Fig. S2 A), suggesting that N-propeptide cleavage does not take place. These results were supported by additional immunoblots detecting the SBP tag (Fig. 4 C). In this instance, full-length procollagen is also evident in WT cells, likely due to the higher affinity of the SBP antibody, but the N-propeptide remains undetectable in the KO cells.

To verify our interpretation of these observations with respect to the N-propeptide, we performed immunoblots using antibodies specifically targeting the N- and C-propeptide domains of pro-α1(I). C-propeptide cleavage was comparable in WT and KO cells (Fig. 4 D), with a band of ~30 kD detected as expected. However, an antibody against the N-propeptide detected a band of ~60 kD in WT cell lysates that was again

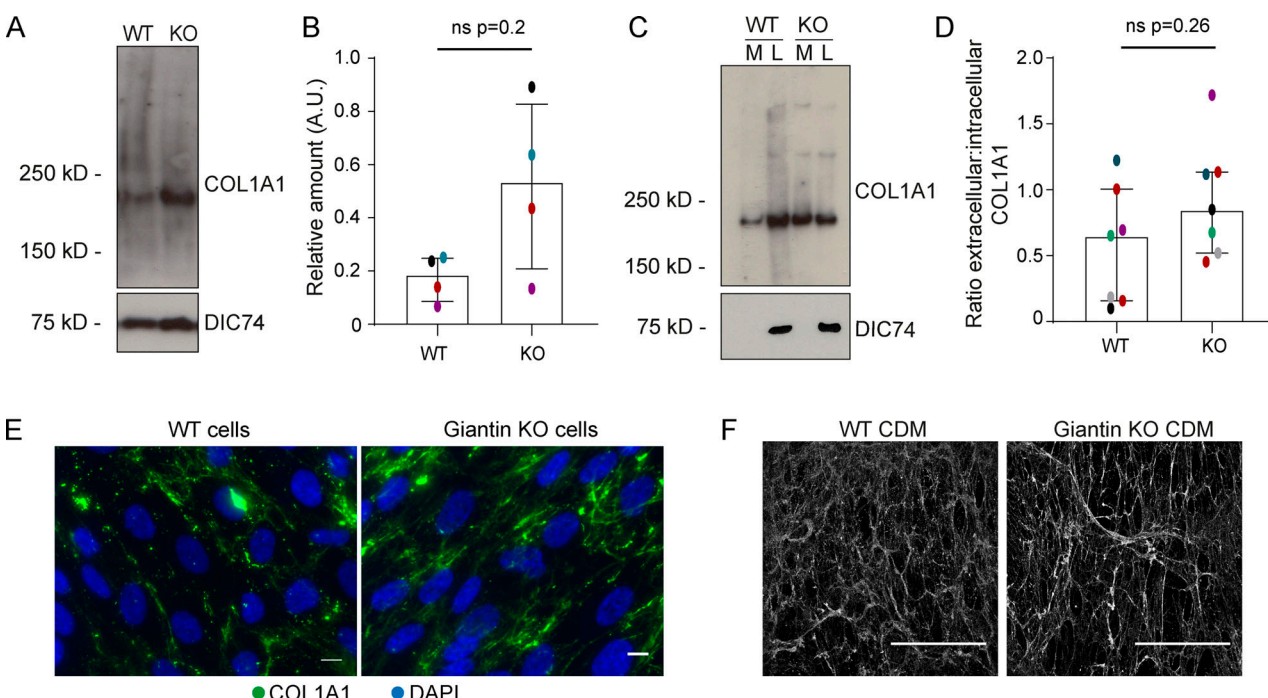

**Figure 2. pro-α1(I) is more abundant in giantin KO RPE1 cells. (A)** Immunoblot of pro-α1(I) and dynein intermediate chain (DIC74, housekeeping) in cell lysates taken from WT and giantin KO cells. **(B)** Densitometry of the semiquantitative ECL immunoblots represented in A. Dots show individual replicates, and each independent experiment is color coded between cell lines. Bars depict median intensity of pro-α1(I) (COL1A1) normalized against DIC74 (n = 4 biological replicates). Error bars = interquartile range. Statistical test: Mann-Whitney U test. **(C)** Immunoblot of media (M) and cell lysate (L) fractions taken from WT and giantin KO cell cultures after 16-h incubation with serum-free medium plus 50 µg/µl ascorbate. **(D)** Ratio of extracellular versus intracellular levels of collagen as measured from secretion assays represented in C. pro-α1(I) (COL1A1) levels measured by densitometry from semiquantitative enhanced chemiluminescent blots and normalized against DIC74 before calculating ratios. Dots show individual replicates, and each independent experiment is color coded between cell lines. Bars depict median and interquartile range (n = 7 biological replicates). Statistical test: Mann-Whitney U test. **(E)** Maximum projections of widefield image stacks showing PFA fixed, unpermeabilized cells immunolabeled for endogenous pro-α1(I) (COL1A1, green). Nuclei are stained with DAPI (blue). Scale bars = 10 µm. **(F)** The cell-derived matrix produced by WT and giantin KO cells imaged as tilescans of confocal z-stacks of antibody-labeled pro-α1(I) presented as maximum projections. Scale bars = 2 µm (E) and 100 µm (F). A.U., arbitrary units.

completely absent in the KO cultures (Fig. 4 E). This N-propeptide band was also detected in the media of WT cells but only following enrichment by immunoprecipitation (Fig. S2 A). It is therefore present but in low quantities. Full-length collagen protein levels were again higher in the KO cells (Fig. 4 F).

We also used the N-propeptide–specific antibody to immunolabel the ECM of WT and KO cells. While this antibody did not label the cell-derived matrix of WT cells, it did label fibrils in the KO cell matrix where it colocalized with the extracellular GFP signal (Fig. 4 G). Together, these data indicate that the presence of GFP in the collagen matrix of KO cells is due to a failure in N-propeptide cleavage.

Secretion of unprocessed procollagen has been reported in the absence of Hsp47 (Ishida et al., 2006); however, Hsp47 levels in the KO cells were normal (Fig. S2, B and C). We also hypothesized that the absence of N-propeptide in the KO cells could be due to its increased degradation, but treatment of KO cells with proteasomal and lysosomal inhibitors did not render it detectable (Fig. S2 D).

**N-propeptide cleavage occurs inside the cell in RPE1 cells**

The detection of free N-propeptide in WT cell lysates is consistent with intracellular cleavage. N-propeptide abundance was also sensitive to proteasomal and lysosomal inhibitors in these

assays, consistent with an intracellular pool (Fig. S2 D). It is, however, also possible that the observed N-propeptide is associated with the cell surface and internalized for degradation. To confirm that processing is indeed an intracellular event, we performed two further biochemical assays on the WT cells stably expressing the GFP-COL1A1.

First, a cell surface trypsin digest was performed to see whether the N-propeptide is in fact at the plasma membrane. As expected, treatment of cell cultures with trypsin for increasing amounts of time resulted in the gradual degradation of EGFR, a known resident of the plasma membrane (Fig. 5 A). Both HSP47, a known resident of the secretory pathway, and free N-propeptide were unaffected. As a control, cells were also briefly permeabilized with digitonin before assay to grant the trypsin access to intracellular proteins. This resulted in the complete degradation of EGFR, including EGFR previously protected in recycling endosomes, whereas soluble cytosolic proteins such as GAPDH and DIC74 were lost during permeabilization. HSP47 and the N-propeptide were again unaffected, suggesting that they remained protected within digitonin-resistant membranes, such as those of the secretory pathway. The N-propeptide is therefore not present on the cell surface or in endosomes.

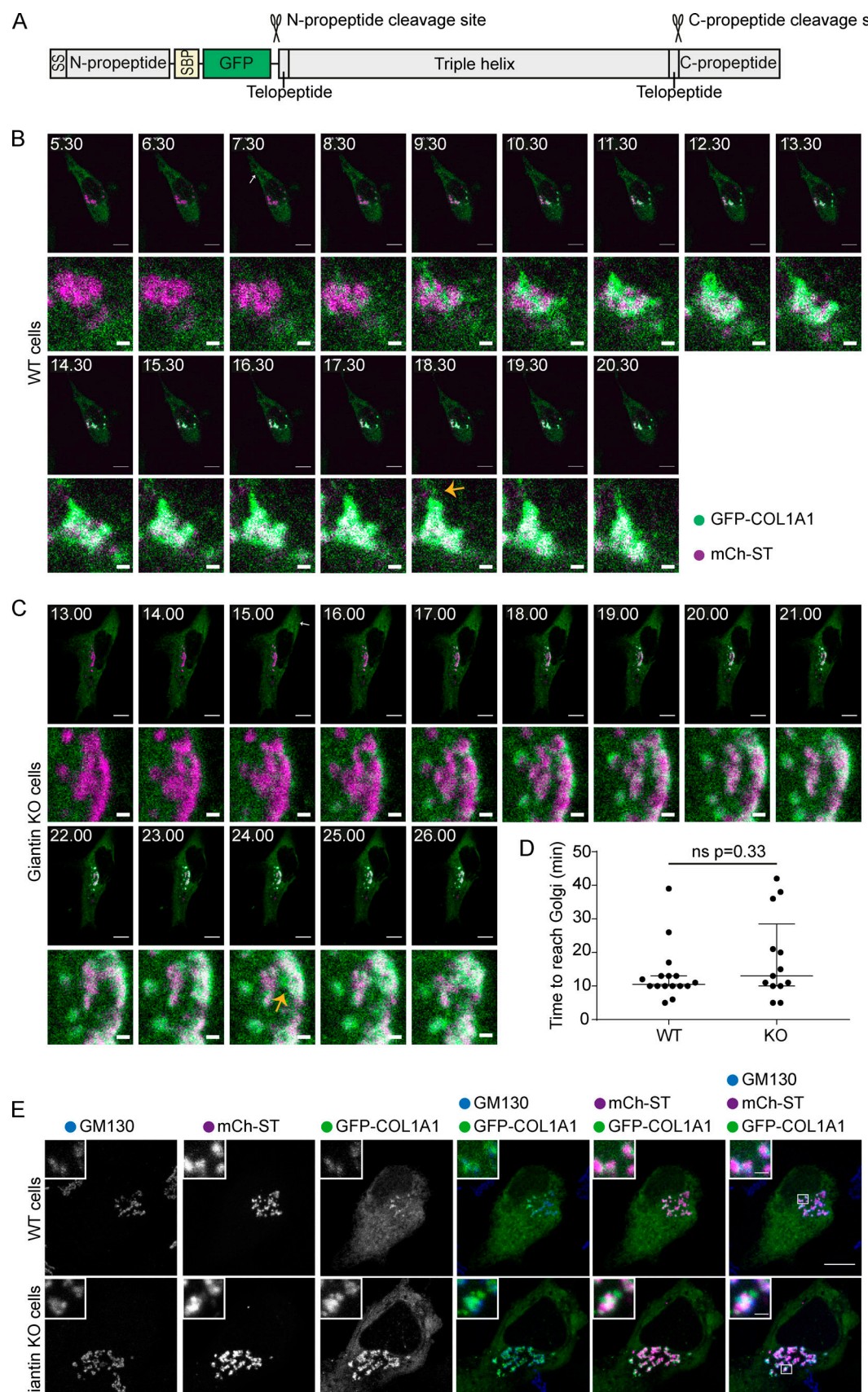

● GFP-COL1A1

● mCh-ST

**Figure 3. ER-Golgi trafficking of pro-α1(I) is unperturbed in giantin KO RPE1 cells. (A)** Schematic of the pro-SBP-GFP-COL1A1 construct used in this study. SS, signal peptide. **(B and C)** Single-plane confocal stills taken from live imaging movies of a WT (B) and giantin KO (C) cell stably expressing GFP-COL1A1

(green) and transiently transfected with mCh-ST (magenta) and an ER RUSH hook. Frame times in the top left corner indicate the time (minutes) after biotin addition, which releases GFP-COL1A1 from the ER hook. Scale bars = 10 µm on overview and 1 µm on zoom. White arrows indicate the appearance of peripheral GFP-COL1A1 punctate. Orange arrows indicate emerging post-Golgi carriers. See also videos. **(D)** Quantification of the time taken for GFP-COL1A1 to first appear at the Golgi in videos represented in B and C. Each dot represents one cell (16 WT cells and 13 giantin KO cells), and videos were collected over seven independent experiments. Bars represent median and interquartile range. Results are not significant using a Mann-Whitney $U$ test. **(E)** Maximum projections of confocal z-stacks of WT and giantin KO cells imaged live as in B and C and then fixed with PFA as soon as the GFP-COL1A1 appeared around the Golgi. Cells were then immunolabeled for the Golgi marker GM130 (blue). Scale bars = 10 µm.

Secreted proteins can be prevented from leaving the Golgi using a temperature block at 20°C. In our second assay, we therefore performed secretion assays at 20°C to see whether procollagen can still be processed if trapped in the ER and Golgi. This would localize cleavage activity to one of these compartments. To ensure we only assayed procollagen affected by the temperature block, we first flushed out all the preexisting procollagen by treating cells with ascorbate to encourage secretion and cycloheximide to prevent synthesis of new procollagen. This removed most of the N-propeptide from the cell layer (Fig. 5 B). The cycloheximide was then washed out to permit expression of new protein, and cells were incubated at 37°C or 20°C overnight in the presence of ascorbate before performing a secretion assay. The accumulation of procollagen in the Golgi at 20°C was confirmed by immunofluorescence (Fig. 5 C). Consistent with intracellular processing, immunoblotting of the assay samples detected free N-propeptide at both temperatures (Fig. 5 B); however, as expected, the N-propeptide was only secreted at 37°C. Altogether, these results are consistent with N-terminal procollagen processing taking place in the early-mid secretory pathway.

### N-terminal processing of type I procollagen is defective in other giantin KO lines

Since the above data were derived from a single clone of *GOLGB1* KO cells, we generated new KO cell lines in which to validate our findings. This time we targeted exon 13 of the *GOLGB1* gene (instead of exon 7) and used the lentiCRISPRv2 system (Sanjana et al., 2014; Shalem et al., 2014) for transfection to vary our approach. We obtained three clones with different indel mutations (Fig. S3 A). Immunoblots with a giantin polyclonal antibody confirmed the loss of full-length protein, although hints of a potential truncated form could be seen in low abundance (Fig. 6 A). Immunofluorescence staining with two different polyclonal antibodies directed against giantin did not detect any protein (Fig. 6 B). As previously described for the original KO line (Stevenson et al., 2017), no gross changes in Golgi morphology or ER exit site abundance were apparent following loss of giantin (Fig. 6 B).

Using the exon 13 mutant clones, we derived new stable cell lines expressing GFP-COL1A1. We also made a new WT GFP-COL1A1 stable line to be sure of the WT phenotype. As with our original clone, we were not able to detect cleaved N-propeptide in either the media or lysate fractions from these new KO cell cultures (Fig. 6 C). We did, however, note some variability in the molecular weight of the procollagen bands detected between clones, perhaps suggesting additional processing defects or deficiencies in post-translational modifications. Imaging of the new KO clones again also confirmed incorporation of the GFP-

tag and the N-propeptide into the extracellular collagen matrix in two of these lines (Fig. 6 D). The phenotypes of the new WT stable cell lines also validated those of the original (Fig. 6, C and D). Overall, these results confirm that N-propeptide processing is defective in the absence of giantin.

Although we found that RPE1 cells do secrete and assemble type I collagen in their matrix, their primary function is to produce nonfibrillar basement membrane. Due to the skeletal phenotypes observed in the animal models, we therefore decided to further test our observations in the more relevant MC3T3 mouse osteoblast precursor cell line. Using CRISPR-Cas9 genome engineering, we generated two mutant lines with gRNAs targeting either exon 2 or exon 8. The latter was genotyped to confirm mutation (Fig. S3 B). Almost a complete loss of giantin was apparent in these clones both by immunoblot and immunofluorescence (Fig. S4, A and B); however, some protein did persist. Again, Golgi morphology was unaffected by the loss of giantin (Fig. S4 B).

Unlike the RPE1 cells, overall levels of expression of full-length procollagen were comparable between the mutants and WT cells in this line (Fig. S4 A). To study procollagen processing, we transiently transfected cells with GFP-COL1A1 and performed a GFP-trap on the media and lysate fractions of WT and mutant cultures (Fig. S4 C). Note, transfection efficiencies were variable between lines but were always low, making analysis difficult. In agreement with the RPE1 cells, immunoblots of the cell culture media showed that the N-propeptide is secreted by WT cells but not by the mutant lines (Fig. S4 C). Small amounts of the N-propeptide were, however, detectable in the KO cell lysates, suggesting some cleavage is taking place. Unfortunately, as in the RPE1 cells, we were unable to detect endogenous free N-propeptide in the MC3T3 cell lysates (Fig. S4 A). Whether this is because it is not present or is in too low abundance is not clear. Overall, the difference in N-propeptide abundance and localization between the WT and giantin mutant MC3T3 is consistent with procollagen N-terminal processing being sensitive to giantin function, even though the outcome manifests slightly differently between systems.

## Discussion

In this study, we demonstrate for the first time that giantin is required for N-terminal, but not C-terminal, processing of type I procollagen. In its absence, procollagen intermediates still bearing the N-propeptide are secreted and incorporated into the ECM. Unlike the C-propeptide, retention of the N-propeptide does not preclude fibril formation (Hulmes et al., 1989; Miyahara et al., 1984; Miyahara et al., 1982; Romanic et al., 1992); thus, it is not surprising that collagen fibrils are formed. We also

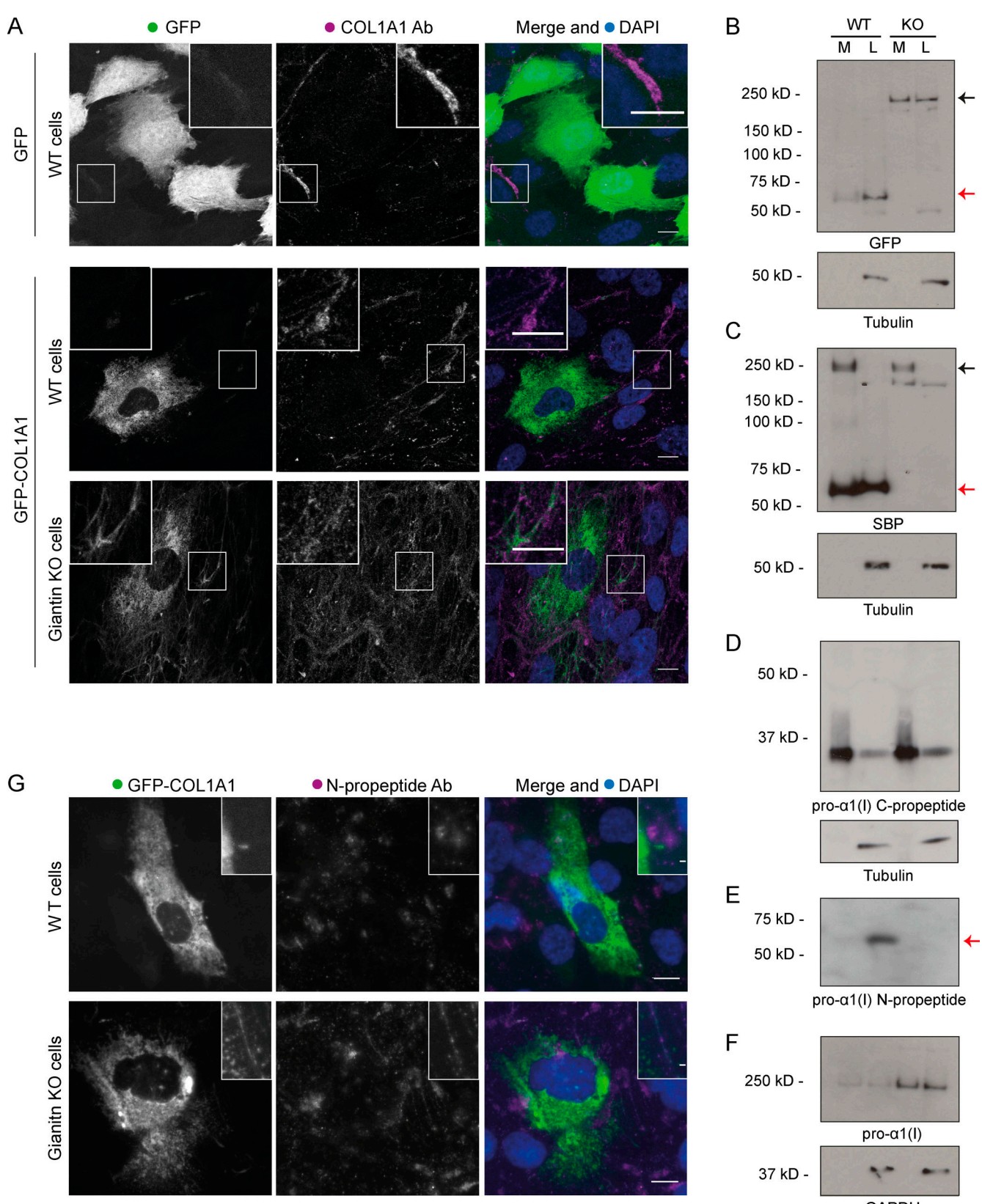

Figure 4. **Procollagen processing is defective in giantin KO RPE1 cells. (A)** Representative images of WT and giantin KO cells expressing GFP-COL1A1 or GFP alone as indicated and immunolabeled for pro-α1(I) (COL1A1, magenta). Cells are not expressing a RUSH hook. Nuclei are stained with DAPI (blue). Images are maximum-intensity projections of widefield z-stacks. Scale bars = 10 μm. **(B–F)** Immunoblots of secretion assays showing the medium (M) and lysate (L) fractions of WT and giantin KO cell cultures immunoblotted for GFP and tubulin (B), SBP and tubulin (C), pro-α1(I) C-propeptide (LF41 antibody) and tubulin (D), pro-α1(I) N-propeptide (LF39 antibody; E), and pro-α1(I) and GAPDH (F). Black arrows indicate full-length procollagen, and red arrows highlight N-propeptide bands. **(G)** Maximum-intensity projection widefield z-stacks of WT and giantin KO cells stably expressing GFP-COL1A1 immunolabeled for the pro-α1(I) N-propeptide (LF39 antibody). Scale bars = 10 μm. Ab, antibody.

Figure 5. **Intracellular procollagen processing in RPE1 cells. (A)** Immunoblots of WT RPE1 cells stably expressing GFP-COL1A1 that were exposed to trypsin for the indicated amount of time before lysis. Lanes 2 and 8 contain lysates from cells that were treated with trypsin + an equal volume of soybean trypsin inhibitor as a negative control. Samples in lanes 1–6 were treated with digitonin for 2 min before trypsin treatment to permeabilize the cells. The DIC74, HSP47, and EGFR blots shown are all from a single membrane, but the scan images were digitally cut in half and the halves swapped to align the correct lanes with the ±digitonin labels in the figure. **(B)** Immunoblots of secretion assays performed on WT RPE1 cells incubated at 20°C or 37°C overnight in the presence of ascorbate. Prior to overnight incubation, cells were treated with cycloheximide (CHX) and ascorbate to flush through pre-existing procollagen. Lane 7 contains a lysate sample taken at the end of this cycloheximide treatment to show the extent to which this removed the procollagen. Lane 1 and 2 samples show media (M) and lysate (L) fractions of a cell culture treated with ascorbate only without the cycloheximide. **(C)** Single-plane widefield images of cells subjected to the experimental conditions of B but fixed after overnight incubation at 20°C or 37°C in the presence of ascorbate. Cells were stained by immunofluoresence for GM130 (magenta) to label the Golgi and DAPI stained (blue) to show the nuclei. Scale bars = 10 µm.

found that the N-propeptide is cleaved inside the cell. Altogether, these data indicate the presence of a giantin-dependent intracellular pathway for procollagen processing.

The most likely cause of the observed processing deficiency in KO cells is a defect in the expression, trafficking, or processing of an N-proteinase. Although we have not yet identified any trafficking defects in our giantin KO cells either generally (Stevenson et al., 2017) or specifically for procollagen, it is possible that other cargoes such as processing enzymes will be affected. To date, five enzymes have been implicated in the N-terminal processing of type I procollagen: ADAMTS2, ADAMTS3, ADAMTS14 (Bekhouche and Colige, 2015; Colige et al.,

1997; Colige et al., 2002; Fernandes et al., 2001), meprin-α, and meprin-β (Broder et al., 2013). Our previous RNA-seq analysis of RPE1 cells (ArrayExpress accession no. E-MTAB-5618) did not detect any ADAMTS2 transcript, even in WT cells; however, ADAMTS3 and ADAMTS14 mRNA was present, and interestingly, their expression was down-regulated following giantin KO (Stevenson et al., 2017). Alternatively, giantin could affect the trafficking of an inhibitor of N-proteinase activity, such as TIMP3. Unfortunately, our attempts to further these lines of inquiry have been unfruitful due to a lack of suitable reagents.

A second possible cause of defective processing is glycosylation deficiency. Indeed, it has been shown that C-proteinase

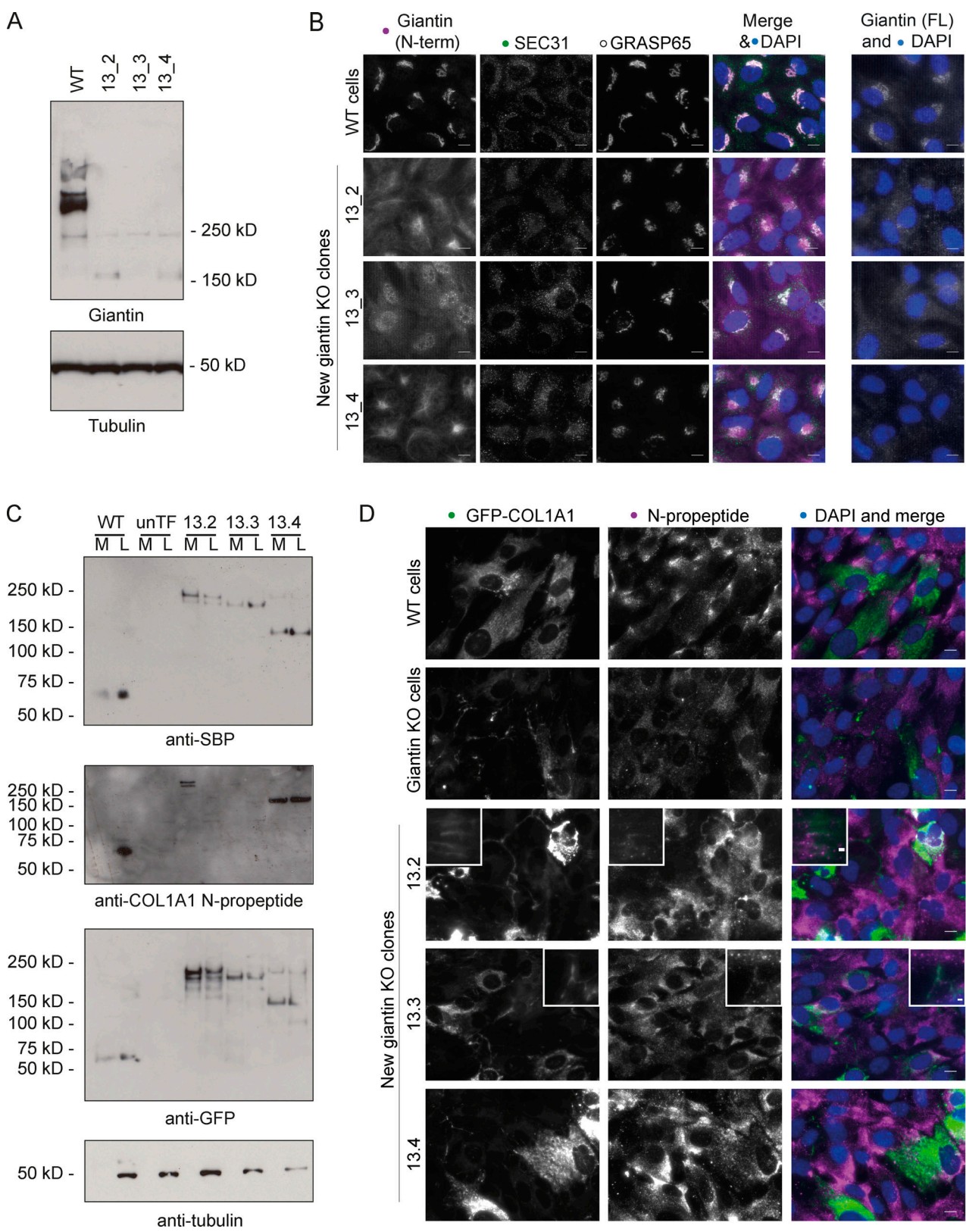

Figure 6. **Validation of processing defects in other giantin mutant RPE1 lines. (A)** Immunoblot of lystates from WT and giantin exon 13 KO RPE1 clones probed for giantin (polyclonal against full-length protein) and tubulin (housekeeping). **(B)** Widefield maximum-projection images of WT and giantin KO RPE1 cell clones immunolabeled for the N-terminus (N-term) of, and full-length (FL) giantin; ER exit sites (Sec31); and the Golgi (GRASP65) as indicated. Scale bar = 10 μm. **(C)** Immunoblots of secretion assays performed on WT and giantin KO clones stably expressing GFP-COL1A1 and untransfected giantin KO cells (unTF). Blots show the medium (M) and lysate (L) fractions of each cell population immunoblotted for GFP, SBP, and pro-α1(I) N-propeptide as indicated. **(D)** Maximum-projection widefield images of GFP signal in giantin KO clones stably expressing GFP-COL1A1 and immunolabeled for pro-α1(I) N-propeptide (magenta). Scale bar = 10 μm.

activity is sensitive to inhibitors of *N*-glycosylation (Duksin et al., 1978). Loss of giantin function affects the enzyme composition of the Golgi and the *O*-glycosylation of specific substrates (Lan et al., 2016; Stevenson et al., 2017). It is thus feasible that at least one of the N-proteinases is incorrectly glycosylated in giantin KO cells in such a way as to impede activity. Alternatively, procollagen itself may be incorrectly glycosylated, preventing its identification as a substrate. We did perform a global analysis of glycoproteins by mass spectrometry, but as predicted from previous results (Stevenson et al., 2017), there were no differences in the major glycosyl chains produced by WT or KO cell lines. This approach did not permit us to determine the glycosylation of specific substrates.

A crucial observation made in this study is that N-propeptide processing occurs, at least in part, inside the cell. More specifically, our data suggest it takes place in the early-mid secretory pathway since Golgi exit is not required for cleavage. There is precedent for the detection of intracellular procollagen processing in both chick and mouse tendons (Canty-Laird et al., 2012; Canty et al., 2004; Humphries et al., 2008). In these studies, processed forms of type I procollagen were detected in a detergent-soluble fraction from tendon explants, consistent with an intracellular pool. Furthermore, inhibition of procollagen secretion by Brefeldin A treatment, which causes the cis- and medial-Golgi machinery to accumulate in the ER, does not prevent N-terminal processing, indicating procollagen can be processed in the early secretory pathway where giantin is located (Canty-Laird et al., 2012). Considering this, together with the data presented here, we conclude that giantin is a crucial component of an intracellular pathway of procollagen processing.

Comparison of our giantin mutant cell lines suggests that processing defects can manifest in different ways. We previously showed that cells alter gene expression to compensate for a lack of giantin, and perhaps the way different systems compensate can affect outcome. This would certainly appear to be the case when comparing phenotypes in animal models. Unlike RPE1 cells, small amounts of N-propeptide were detected in the KO MC3T3 cells. Whether this N-propeptide pool is in the secretory pathway and is a bona fide product of a processing event, or whether it is present in the degradative pathway as a byproduct of procollagen turnover after overexpression was not ascertained. The fact that it is not secreted perhaps suggests the latter is more likely. These cell populations also retain some giantin, and so the phenotype may not be as complete as in the RPE1 cells. Similarly, in these experiments, a human recombinant procollagen was expressed in a mouse cell line, which may affect how the procollagen is recognized and processed.

We found that *COL1A1* promoter activity and mRNA levels are reduced in both *golgb1*[X3078/X3078] mutant zebrafish and KO cells, respectively (Stevenson et al., 2017). This contrasts with protein levels, which are more abundant in vitro, at least in the RPE1 cells, which have a more complete processing defect than the MC3T3 cells. Previous studies have suggested that the pro-α1(I) N-propeptide is capable of acting as a negative regulator of pro-α1(I) synthesis (Hörlein et al., 1981; Oganesian et al., 2006; Paglia et al., 1979; Wiestner et al., 1979). Importantly, the point of

inhibition occurs during translational chain elongation or termination (Hörlein et al., 1981; Oganesian et al., 2006; Paglia et al., 1979). Thus, we speculate that despite the reduced abundance of mRNA transcript, translation of pro-α1(I) is more efficient in giantin KO RPE1 cells in the absence of inhibitory N-propeptide. Consistent with reports that only endogenously expressed N-propeptide can inhibit translation, we were unable to rescue our phenotypes by supplementing media with recombinant type I procollagen N-propeptide.

Human patients with mutations affecting N-terminal processing of type I procollagen suffer from Ehlers-Danlos syndrome (EDS) type VII, a disease characterized by joint hypermobility, skin hyperextensibility, and dislocations (arthrochalasia type, formally type VIIA and VIIB) or skin fragility (type VIIC/dermatosparatic type; Byers et al., 1997; Colige et al., 1999; Giunta et al., 1999; Smith et al., 1992). Collagen fibers in dermatosparatic-type patients appear hieroglyphic in cross section and have irregular contours in type VIIB patients (Malfait et al., 2013; Van Damme et al., 2016). ADAMTS2-deficient mice also have thinner collagen fibrils in the skin (Li et al., 2001). We primarily examined collagen fibril structure in the bones of the *golgb1*[X3078/X3078] fish; however, skin collagen was also present in these samples, and we did not see any structural changes. The fact that the fish do not recapitulate an EDS phenotype could suggest that the pathways affected by giantin loss may be ADAMTS2 independent, especially given the sensitivity of the RPE1 cells, which do not express ADAMTS2.

As previously described, *golgb1*[X3078/X3078] mutant zebrafish show developmental growth delay, elongated craniofacial structures, and ectopic mineralization of soft tissue (Bergen et al., 2017; Stevenson et al., 2017). Here, we additionally report that mutant adult caudal fins accumulate significant numbers of fractures and show defects in the maturation of newly deposited matrix during fracture repair. Our observations of increased calcein labeling in mutant fish fractures is consistent with either increased osteoid formation or hypermineralization. We previously reported that expression of the glycosyltransferase GALNT3 is reduced in both *GOLGB1* KO cells and zebrafish (Stevenson et al., 2017). GALNT3 is required to glycosylate and activate FGF23, which is an inhibitor of bone mineralization (Wang et al., 2008). The observed defects in fracture repair could therefore be the result of FGF23 deficiency. Alternatively, or perhaps additionally, collagen N-propeptides have been linked to the differentiation and apoptosis of osteoblasts (Oganesian et al., 2006) and osteoclasts (Hayashi et al., 2011), respectively. We have not been able to directly demonstrate that the procollagen processing defects observed in cells translate to fish; however, it is interesting to speculate that processing defects may affect the balance of bone synthesis and resorption in the mutant fractures. The significant impact of the loss of giantin on mineralization likely explains why mutant animals primarily present with skeletal defects and why other type I collagen-rich tissues, such as skin, are less affected.

The observed increased incidence of fractures in mutants could result from an increased frequency of fractures in brittle bones or an accumulation of fractures due to slow healing. Although fracture repair is delayed in the mutants, our data show

that the described differences between WT and mutant fractures largely resolve within 3 wk. This seems insufficient to account for the number of spontaneous fractures that accumulate, given that callouses can persist for months overall (Geurtzen et al., 2014). Combined with the clustered nature of a significant number of fractures, we believe it likely that the bones are more brittle. This would be consistent with a defect in type I collagen deposition, as mutations in the *COL1A1* gene are associated with osteogenesis imperfecta in both humans (Glorieux, 2008) and zebrafish (Fisher et al., 2003; Gistelinck et al., 2018). Some N-propeptide processing deficiencies can cause disease with characteristics of both osteogenesis imperfecta and EDS in humans (Cabral et al., 2005; Makareeva et al., 2006). Giantin KO also affects the abundance of other matrix proteins important for bone quality (Stevenson et al., 2017).

A limitation of our study is that we have been unable to link the processing defect seen in cell culture with the phenotypes observed in zebrafish mutants. A simple explanation for this is that extracellular processing in vivo compensates and precludes detection of this processing defect. This would be consistent with the mild impact on matrix organization observed. In cases of EDS dermatosparaxis type and arthrochalasia type, mutations in ADAMTS2 or the N-terminus of pro-α1(I)/pro-α2(I), respectively, directly impact N-propeptide cleavage regardless of cellular location or tissue environment. We propose that giantin acts more indirectly to regulate an intracellular processing pathway, which works in concert with other complementary mechanisms, such as extracellular cleavage, for coordinated and efficient processing under different conditions. This raises the intriguing possibility of specific pools of collagen that require intracellular processing for specific functions, such as fracture repair. It is also possible that the observed phenotypes have a different underlying mechanism. RNA-seq data showed that the expression of many matrix components is altered following loss of giantin (Stevenson et al., 2017), which must affect matrix quality more globally. Giantin also impacts ciliary function, which would compound phenotypes (Asante et al., 2013; Bergen et al., 2017). Further study will be needed to define which, if any, phenotypes seen in zebrafish are a direct result of defective N-terminal processing of type I procollagen.

## Materials and methods
### Zebrafish husbandry and genetic lines
London AB zebrafish were maintained using standard conditions (Aleström et al., 2020). Ethical approval was obtained (University of Bristol Ethical Review Committee), and experiments were performed under Home Office Project License number 30/3804. The *golgb1$^{bsl077}$* (henceforth *golgb1$^{X3078}$*) allele is described in Bergen et al. (2017). The *golgb1$^{X3078}$* allele was crossed with fish carrying the *Tg(col1a1a:EGFP)$^{zf195tg}$* (henceforth *col1a1a:GFP*) transgenic reporter of the 1.4-kb *col1a1a* promoter region (Kague et al., 2012). The GFP-positive *golgb1$^{wt/X3078}$* + *col1a1a:GFP* individuals were in-crossed to generate homozygotes. In all experiments, homozygotes were compared with siblings.

### Fin fracture assays
Fin fractures were performed as previously described (Geurtzen et al., 2014). Briefly, 7-mo-old adult fish were anesthetized with MS222 (Sigma-Aldrich) and moved into a petri dish with the head placed on a bed of tissue soaked in anesthetic and the tail splayed on the dish for imaging. Caudal fins were imaged before injury, and any pre-existing fractures were counted. Fractures were induced by pressing on an individual segment of bone in the tail rays with the end of a semi-flexible plastic pipette tip. Four fractures per fish were introduced in non-pigmented sections of fin in the distal third of the tail, avoiding the four most dorsal and ventral rays. Fish were then reimaged live at various time points after injury.

### Live staining of bone
To visualize bone repair in live zebrafish, ARS and calcein green stain were used. ARS solution contained 74 µM Alizarin Powder (Sigma-Aldrich) and 5 mM Hepes dissolved in Danieau's buffer. Calcein green staining solution contained 40 µM calcein powder (Sigma-Aldrich) dissolved in Danieau's buffer and was pH calibrated to pH 7.4. Live fish were immersed in either ARS or calcein green for 2 h, then immersed in fresh system water for 15 min before imaging to clear excess stain from the fish.

Fluorescence intensity at fracture sites was measured using ImageJ; lines were drawn along the fracture callous length, and then the line width was expanded to encompass the whole fracture. The mean intensity was then measured across this area, before using the same region of interest to measure the mean intensity of adjacent, uninjured hemirays. Fracture intensity was then normalized against the healthy bone to account for differences in baseline expression caused by reporter integration number.

### TEM of zebrafish caudal fin
For caudal fin lysates, two female and two male 8-mo-old *golgb1$^{wt/X3078}$;col1a1a:GFP$^+$* and *golgb1$^{X3078/X3078}$;col1a1a:GFP$^+$* fish were anesthetized in MS222 and placed in a petri dish with the head on a bed of tissue soaked in anesthetic and the tail splayed on the dish. With a fresh scalpel, the distal third of the caudal fin was amputated. After recovery in fresh water, the fish were then returned to their tanks for 7 d. The regenerated fin tissue was then amputated as above, cutting as close to the original dissection line as possible. The excised tissue was then cut in half to separate the immature tip of the fin and the more mature tissue. The tissue was fixed in 2.5% glutaraldehyde, 4% PFA, 0.05 M cacodylate, 1 mM MgSO$_4$, and 1% sucrose for 4 h at room temperature with gentle agitation. Fins were washed 3× in 0.1 M cacodylate buffer and incubated in decalcification buffer (0.1 M EDTA, 0.1 M cacodylate buffer, and 2.5% glutaraldehyde, pH 7.2) at 4°C for 2 wk, refreshing the buffer after 1 wk. Fins were then postfixed in a freshly made mixture of 2% osmium tetroxide and 1.5% potassium ferrocyanide in 0.1 M cacodylate buffer for 1 h. Fix was rinsed off with 3× 5-min washes with water and then incubated with 1% tannic acid in 0.1 M cacodylate buffer for 2 h. Tissue was then washed again as before incubating in 2% osmium tetroxide in distilled water for 40 min at room temperature. After a further three washes in distilled water, tissue was

stained with 1% uranyl acetate overnight before washing again. The fins were then subjected to a graded dehydration, incubating them in 50%, 70%, and 90% ethanol, then 100% ethanol four times for 10 min each. This was followed by a 15-min incubation in propylene oxide. Samples were gradually infiltrated with graded TAAB 812 premix kit–Hard-1 (TAAB) in acetone at room temperature using the following series: 25% TAAB overnight, 50% TAAB through the day, 75% TAAB overnight, and 100% TAAB for 6 h. Samples were then embedded in fresh 100% TAAB 812 Hard in a labeled mold at 60°C for 48 h. The more mature pieces of regenerated fins were then sectioned and imaged using an FEI Tecnai 12 BioTwin transmission electron microscope.

### Collecting zebrafish lysates

To collect larval lysates, *golgb1*$^{X3078/X3078}$*;col1a1a:GFP* homozygotes were out-crossed with EEK strain WT fish to generate heterozygous larvae and also in-crossed for homozygotes. At 5 dpf, larvae were killed by overdose with MS222, and 20 larvae were transferred to an Eppendorf for lysis. Larvae were incubated in Ringers solution (116 mM NaCl, 2.9 mM KCl, 1.8 mM CaCl$_2$, and 5 mM Hepes, pH 7.2) for 30 min and then pipetted up and down to remove yolk. Tubes were centrifuged at 13,000 ×*g* for 5 min, the supernatant was discarded, and 100 µl RIPA buffer + 10% protease inhibitor cocktail was added per tube. Larvae were homogenized using a pellet pestle to agitate tissue for 1 min per sample and then left on ice for 30 min, mixing halfway by flicking the tube. Lysates were then centrifuged for 30 min at 4°C and 13,000 ×*g*. The supernatant was analyzed by SDS-PAGE, and the pellet was discarded.

For caudal fin lysates, 13-mo-old *golgb*$^{wt//X3078}$*–col1a1a:GFP* and *golgb1*$^{x3078/X3078}$*;col1a1a:GFP* were anesthetized, and the distal third of the caudal fin was removed as above for TEM and placed into 200 µl ice-cold PBS. After recovery in fresh water, the fish were returned to their tanks for 4 d. The fish were then culled by overdose in MS222, and the caudal fin was cut again, this time dissecting ~2 mm proximal to the original cut site to collect the regenerated tissue plus a small amount of mature tissue for lysis.

The dissected fins were macerated in the PBS using scissors, then spun at 13,000 ×*g* for 4 min at room temperature. Tubes were put on ice, the PBS was removed, and 100 µl RIPA buffer + 10% protease inhibitor cocktail was added per tube. Tissue was homogenized for 40 s with a pellet pestle and then incubated for 10 min at 4°C on a shaker. Samples were spun at 13,000 ×*g* for 10 min at 4°C, and the supernatant was collected for SDS-PAGE conducted as detailed below.

### Cell culture and genome engineering

hTERT-RPE1 cells (American Type Culture Collection) were grown in DMEM-F12 (Life Technologies) supplemented with 10% decomplemented FCS (Gibco). Cell lines were not authenticated after purchase other than confirming the absence of mycoplasma contamination. The main *GOLGB1* KO hTERT-RPE1 cell line used is described in Stevenson et al. (2017). MC3T3 cells (MC3T3-E1 ECACC 99072810) were grown in MEMα (nucleosides, no ascorbate; Life Technologies; catalog #A1049001) supplemented with 10% decomplemented FCS (Gibco), 100 U/ml penicillin, and 100 µg/ml streptomycin.

New hTERT-RPE1 *GOLGB1* KO cell lines with mutations in exon 13 were generated using the lentiCRISPRv2 system (lenti-CRISPR v2 was a gift from Feng Zhang [Massachusetts Institute of Technology, Cambridge, MA]; Addgene plasmid #52961; http://n2t.net/addgene:52961; Research Resource Identifier [RRID]: Addgene 52961). gRNA sequences were designed using Benchling (http://benchling.com) and inserted into the lenti-CRISPRv2 construct as described in Sanjana et al. (2014) and Shalem et al. (2014) to be coexpressed with Cas9. Target sequences used were 5′-CCACCGGGAAGCCTTAACCTCCCGCA-3′ and 5′-AAACTGCGGGAGGTTAAGGCTTCCC-3′. The cloned plasmid was packaged into lentivirus using Lenti X Packaging Single Shots (Takara; #631275) according to the manufacturer's instructions. After harvest, 1 ml virus supernatant was added to 80% confluent RPE1 cells in a 6-cm dish (after removal of growth media) in the presence of 8 µg/ml polybrene. Virus was incubated for 1 h before adding back fresh growth medium supplemented with 8 µg/ml polybrene. Transfection medium was replaced after 24 h. Cells were passaged 24 h later in 20 µg/ml puromycin to select transfected cells. After 7 d, individual cells were FACS sorted into a 96-well plate to grow up clones and screen for *GOLGB1* KO by immunofluorescence and immunoblot. To identify the mutations, genomic DNA was extracted from each clone using the Purelink genomic DNA mini kit (Invitrogen), and the region targeted by the gRNAs was amplified by PCR (primers: forward, 5′-GCTGGCAGCTGAAGAGCAATT CCA-3′, and reverse, 5′-GTTGAGTGTGATGCTGTTCTGTGGCT-3′). PCR products were cloned into the pGEM-T Easy vector according to the manufacturer's instructions (Promega) and sequenced using predesigned primers against the T7 promoter (MWG Eurofins).

Stable cell lines were made by lentiviral transduction as described above using the Lenti X Packaging Single Shots (Takara; #631275) in combination with the lentiviral vector pLVXpuro-proSBP–GFP-COL1A1 (Addgene; #110726) described in McCaughey et al. (2019). To ensure only low levels of overexpression, all cell lines were FACs sorted. GFP-only stable cells are described in Asante et al. (2014).

Transient transfections were performed using Lipofectamine 2000 according to the manufacturer's instructions using 2 µg DNA for a 35-mm well (Invitrogen; catalog #11668027). Plasmid: Str-KDEL-IRES-ST-mCherry is described in McCaughey et al. (2019; Addgene; #110727).

The ALT-R CRISPR-Cas9 system from IDT was used to generate giantin KO MC3T3 cell lines. Guide RNA1 targeting exon 8 (KO1) was designed using Benchling (5′-GGAAAAGGTAGAACT CGAAG-3′), and gRNA2 (KO2) was ordered as a predesigned guide from IDT targeting exon 2 (5′-AATAATGGAATCCACGCA AG-3′ Mm.Cas9.GOLGB1.1.AB). The gRNAs and Cas9 RNPs were assembled and transfected in a 96-well plate using Cas9 working buffer (20 mM Hepes and 150 mM KCI, pH 7.5), CRISPRMAX transfection reagent (Invitrogen; CMAX00008), and 20,000 cells/well according to the IDT published protocols. Cells were then expanded, and single cells were sorted into a 96-well dish to generate clones. Again, these were expanded, and giantin KO clones were identified by immunofluorescence and immunoblotting. From 275 cells seeded per gRNA, one KO was obtained

per target. Genetic mutations were determined as for RPE1 cells using primers 5′-GCCTGCCTTCCTTCCTTTAC-3′ and 5′-GCG GAACCAAGCAACAATAC-3′ for exon 2 mutations and primers 5′-AGTGTTTCAGTGTGCTCCC-3′ and 5′-GTCTTCTCCATCTCT GTC-3′ for exon 8 mutations.

## Antibodies

Rabbit anti-COL1A1 (Novus; catalog #NB600-408; RRID: AB_10000511), mouse anti-DIC74 (Millipore; catalog #MAB1618; RRID:AB_2246059), mouse anti-HSP47 (Enzo Life Sciences; catalog #ADI-SPA-470; RRID:AB_10618557), mouse anti-GM130 (BD Biosciences; catalog #610823; RRID:AB_398142), rabbit anti-giantin (N-terminus; BioLegend; catalog #924302; RRID: AB_2565451), rabbit anti-giantin (polyclonal antibodies raised against full-length giantin originally from Prof. Manfred Renz [Karlsruhe, Germany], a gift to us from Martin Lowe, University of Manchester, Manchester, UK; Lowe et al., 2004), mouse anti-SBP clone 20 (Millipore; catalog #MAB10764; RRID: AB_10631872), rabbit polyclonal anti–N- and C-propeptide of COL1A1 (LF-39 and LF-41, respectively; both gifts from Larry Fisher, National Institutes of Health, Bethesda, MD; Fisher et al., 1995), mouse monoclonal anti-GFP (Covance; catalog #MMS-118P-500; RRID:AB_291290), sheep polyclonal anti-GRASP65 (gift from Jon Lane, University of Bristol, Bristol, UK), tubulin (Sigma-Aldrich; catalog #T5168; RRID:AB_477579), mouse monoclonal anti-Sec31A (BD Biosciences; catalog #612350; RRID: AB_399716), and mouse p62 (Novus; catalog #H00008878-D01P; RRID:AB_1504204). Secondary HRP-conjugated antibodies were obtained from Jackson ImmunoResearch. Alexa Fluor–conjugated secondaries were obtained from Invitrogen.

## Immunofluorescence

Cells were grown on autoclaved coverslips (0.17-mm thickness; Thermo Fisher Scientific; #1.5) before fixation with 4% PFA/PBS for 10 min, permeablization with 1% Triton X-100/PBS for 10 min, and blocking with 3% BSA/PBS for 30 min. Sequential incubations with primary and secondary antibodies diluted in block were performed for 1 h, washing in PBS in between. Nuclei were labeled with DAPI (Life Technologies; D1306) for 3 min. Coverslips were mounted in MOWIOL 4–88 (Merck-Millipore). All steps were conducted at room temperature and in the dark after secondary antibody addition. When labeling for ECM, the Triton permeability step was excluded from the protocol.

For widefield microscopy, an Olympus IX-71 inverted microscope was used with a 60× 1.42 NA oil-immersion lens, Exfo Excite xenon lamp illumination, and single-pass excitation, emission, and multipass dichroic (Semrock) filters. Images were captured on an Orca-ER charge-coupled device (Hamamatsu). The system was controlled using Volocity (PerkinElmer; v.5.4.1). Chromatic shifts in images were registration corrected using TetraSpeck fluorescent beads (Thermo Fisher Scientific). Fixed-cell imaging by confocal microscopy was performed using a Leica SP5II system at 1024 × 1024 x-y resolution with a 63× HCX PL APO CS oil immersion objective, photomultiplier tube detectors, and LAS X software. Tile scans were performed to capture larger areas of the cell-derived matrix using a Märzhäuser scanning stage. On both systems, image stacks were taken with Δz of 0.2 µm, and unless indicated, maximum projections are shown. Image processing was performed using ImageJ software (Schindelin et al., 2012).

## Preparation of the cell-derived matrix

To prepare the cell-derived matrix, cells were grown for 3 d on coverslips until they reached confluence, and then 50 µg/ml L-ascorbic acid-2-phosphate (Sigma-Aldrich) was added to cultures. Cells were left for a further 7 d. To prepare samples, cells were washed in PBS and extracted using prewarmed extraction buffer (20 mM $NH_4OH$ and 0.5% Triton X-100 in PBS; 3 ml per 6-cm plate for 2 min). After three water washes, residual DNA was digested with 10 µg/ml DNase I (Roche) for 30 min at 37°C, and then extracts were washed again. The matrix was fixed in 4% PFA and stained as above.

## Live imaging

Procollagen trafficking was analyzed using the RUSH assay (Boncompain et al., 2012). Cells stably expressing pro-SBP-GFP-COL1A1 were grown on 35-mm MatTek glass-bottomed dishes and transfected with Str-KDEL-IRES-ST-mCherry (McCaughey et al., 2019) 24 h before imaging. Cells were always confluent at time of imaging.

To image, growth media was replaced with 1 ml prewarmed Fluorobrite DMEM (Thermo-Fisher Scientific; A1896701), and the dishes were mounted on a Leica SP8 confocal laser scanning microscope system with a 63× HC OL APO CS2 1.42 NA glycerol lens and an environmental chamber at 37°C with $CO_2$ enrichment. The microscope was controlled with Leica LAS X software. Once GFP- and mCh-positive cells were identified, 1 ml Fluorobrite containing 800 µM biotin and 1 mg/ml L-ascorbic acid-2-phosphate (Sigma-Aldrich; A92902) was added to the dish (final concentration 400 µM biotin and 500 µg/ml, respectively) to induce procollagen folding and synchronous release of the pro-SBP-GFP-COL1A1 from the ER hook. This is T = 0. Cells were then imaged live, taking a single confocal plane image every 20 s. Fluorophores were excited using a 65-mW Argon and a 20-mW solid-state yellow laser and detected using hybrid gallium arsenide phosphide detectors. Imaging conditions for movies were 1,024 × 1,024 x-y resolution, scanning speed 600 Hz, 2× zoom, three-line average, and imaging sequentially between frames.

To immediately fix during live imaging, 2 ml 8% PFA was added to the 2 ml Fluorobrite already present at the desired time point. Cells were fixed for 10 min, and the above immunofluorescence protocol was applied.

## Secretion assays, immunoprecipitations (IPs), and cell lysates

To perform secretion assays, medium was aspirated from confluent cells and replaced with a minimal volume of serum-free DMEM-F12 supplemented with 50 µg/ml L-ascorbic acid-2-phosphate (Sigma-Aldrich; A92902). Cells were left overnight at 37°C and 5% $CO_2$. Medium was then collected, and the cell layer was rinsed with PBS, lysed in RIPA buffer (50 mM Tris-HCl, pH 7.5, 300 mM NaCl, 2% Triton X-100, 1% deoxycholate, 0.1% SDS, and 1 mM EDTA) for 30 min rocking on ice and collected without scraping. Media and lysate fractions were spun at 13,000 ×$g$ at

4°C for 10 min and the supernatants used for SDS-PAGE. For collection of lysates in other experiments, cells were scraped in RIPA buffer, collected in tubes, and incubated on a rotator at 4°C for 30 min. Bicinchoninic acid assays were performed where necessary to determine protein concentration according to the manufacturer's instructions (Bio-Rad). Samples were boiled in 1× lithium dodecyl sulphate sample buffer (Life Technologies; NP007) containing sample-reducing agent (Life Technologies; NP007) for 10 min at 95°C and then analyzed by SDS-PAGE as below.

For inhibitor experiments, confluent cells were treated with 1 ml serum-free medium containing either DMSO (control), 10 μM MG132, or 200 nM bafilomycin overnight before media and lysate collection.

For GFP-trap immunoprecipitation, confluent cells were serum starved for 24 h in the presence of 50 μg/μl L-ascorbic acid-2-phosphate (Sigma-Aldrich; A92902). At the time of assay, medium was collected and spun at 2,700 ×g for 5 min to remove dead cells. The cell layer was washed twice with ice-cold PBS and lysed in ice-cold buffer (10 mM Tris-HCl, pH 7.4, 50 mM NaCl, 0.5 mM EDTA, 1.0% Igepal CA-630, 1 mM PMSF, and 1× Protease inhibitor cocktail [Millipore; 539137]) for 30 min with agitation on a rotator. Lysate supernatant was collected by centrifuging at 20,000 ×g for 10 min, and 66 μl was removed to be used as an "input" reference sample. Media and lysate supernatants were then incubated with equilibrated GFP nano-trap beads (Chromotek) on a rotator for 2 h. Beads were pelleted by centrifugation at 2000 ×g for 2 min. The supernatant was removed, but 66 μl was kept as an "unbound" reference sample. The beads were then washed three times with 500 μl dilution buffer (10 mM Tris-HCl, pH 7.4, 50 mM NaCl, 0.5 mM EDTA, 1 mM PMSF, and 1× Protease inhibitor cocktail), pelleting them each time by centrifugation at 2000 ×g for 2 min. All steps to this point were performed on ice/at 4°C. Finally, beads were resuspended in 88 μl 1× LDS sample buffer (Life Technologies; NP007) containing sample reducing agent (Life Technologies) Inbound and unbound samples were mixed with 22 μl 4× LDS sample buffer + reducing agent (Life Technologies). Samples were boiled at 95°C for 10 min and analyzed by SDS-PAGE as outlined below.

### Trypsin digest assay
Confluent RPE1 cells were treated with growth media + 50 μg/ml ascorbic acid for 1.5 h at 37°C/5% $CO_2$ to stimulate procollagen trafficking and then incubated in PBS ± 30 μg/ml digitonin for 2 min at room temperature to permeabilize the cells. Cells were washed twice in PBS and twice in sucrose buffer (0.3 M sucrose, 0.1 M KCl, 2.5 mM $MgCl_2$, 1 mM sodium-free EDTA, and 10 mM Pipes, pH 6.8) before incubating with 0.05% trypsin-EDTA for the indicated time at 37°C/5% $CO_2$. As a control, one well was incubated with a 1:1 ratio of 0.05% trypsin and 2 mg.ml$^{-1}$ soybean trypsin inhibitor (Life Technologies; 17075029; dissolved in PBS). To end the digest, an equal volume of 2 mg/ml soybean trypsin inhibitor was added to the well and incubated for 10 min at room temperature. Dishes were scraped, and the cells were collected and spun at 2,000 ×g for 5 min at 4°C. The supernatant was removed, and the pellet was washed with 500 μl 2 mg/ml

soybean trypsin inhibitor before repeating the spin and removing the supernatant. The cell pellet was lysed in 85 μl RIPA buffer containing proteinase inhibitor cocktail (Millipore; 539137) for 30 min on rotator at 4°C. Lysates were spun at 13,000 ×g/4°C for 10 min, and the supernatant was collected for SDS-PAGE as below.

### Procollagen processing at 20°C
Confluent WT and giantin KO RPE1 cells stably expressing pro-SBP-GFP-COL1A1 were incubated with 50 μg/ml L-ascorbic acid-2-phosphate (Sigma-Aldrich; A92902) ± 100 μg/ml cycloheximide for 16 h to promote secretion of existing procollagen and prevent synthesis of new procollagen, respectively, to remove preexisting protein. As a control to test the efficiency of the cycloheximide treatment, one cycloheximide-treated sample was collected after this incubation. Remaining cells were rinsed 3× with PBS and once with growth media before incubating with $CO_2$ independent media + 50 μg/ml ascorbic acid overnight at either 20°C or 37°C/5% $CO_2$. Cells were then fixed and stained or used in a secretion assay as above—including the control that lacked cycloheximide treatment.

### SDS-PAGE
Secretion assays, IP, and lysate samples were separated by SDS-PAGE on precast 3–8% tris-acetate and 4–12% bis-tris acrylamide gels (Invitrogen) followed by transfer to 0.2 μm nitrocellulose membranes (Amersham). Membranes were blocked in 5% milk 0.05% Tween (Sigma-Aldrich) in Tris-buffered saline for at least 1 h. Primary and HRP-conjugated secondary antibodies were diluted in block and sequentially incubated with the membrane for 2–16 h each. HRP was detected by enhanced chemiluminescence (Promega ECL; GE Healthcare film).

### Statistical analyses
Measurements of images and immunoblots were performed using ImageJ. Statistical analyses were performed using GraphPad Prism 7.00. The tests used, $n$ numbers, and sample sizes are indicated in the figure legends; P values where significant (P < 0.05) are shown on the figures. All experiments were analyzed with nonparametric tests as data were assumed not to be normally distributed. Sample sizes were chosen based on previous similar experimental outcomes. No samples were excluded.

### Online supplemental material
Fig. S1 shows characterization of fractures, collagen expression, and collagen fibril morphology in *golgb1* HET and HOM zebrafish. Fig. S2 contains additional experimental data supporting the absence of N-terminal procollagen processing in the giantin KO cells. Fig. S3 details the genetic changes present in the newly generated giantin KO RPE1 and MC3T3 cells. Fig. S4 shows data confirming the loss of giantin and absence of procollagen type I processing in the newly generated giantin KO MC3T3 cells. Video 1 and Video 2 illustrate synchronized procollagen trafficking in WT (Video 1) and giantin KO (Video 2) RPE1 cells using the RUSH trafficking system.

## Acknowledgments

We thank Franck Perez and Gaelle Boncompain (Institut Curie, France) for sharing the RUSH system with us, Helen Dawe (University of Exeter, UK) and Stuart Haslam (Imperial College London, UK) for their technical help and advice, Andrew Herman (University of Bristol flow cytometry facility) and Kate Heesom (University of Bristol proteomics facility) for their work on this project, Lucy McGowan for help with zebrafish experiments, and Alain Colige (University of Liege, Liege, Belgium) for his help and advice and for providing an ADAMTS2 construct. We also thank the Medical Research Council and Wolfson Foundation for establishing the Wolfson Bioimaging Facility.

This work was funded by the UK Research and Innovation–Medical Research Council (MR/P000177/1), Versus Arthritis (21937 and 22044), and UK Research and Innovation–Biotechnology and Biological Sciences Research Council (BB/T001984/1). Confocal microscopy was also supported by UK Research and Innovation–Biotechnology and Biological Sciences Research Council (BB/L014181/1) and electron microscopy by the Wellcome Trust (110126/Z/15/Z).

The authors declare no competing financial interests.

Author contributions: Conceptualization: D.J. Stephens and N.L. Stevenson. Formal analysis: N.L. Stevenson. Funding acquisition and supervision: D.J. Stephens, C.L. Hammond, and K.E. Kadler. Investigation and methodology: N.L. Stevenson, Y. Lu, M.E. Prada-Sanchez, and D.J.M. Bergen. Project administration: D.J. Stephens. Visualization: N.L. Stevenson. Writing – original draft: N.L. Stevenson and D.J. Stephens. Writing – review and editing: N.L. Stevenson, D.J.M. Bergen, D.J. Stephens, C.L. Hammond, and K.E. Kadler. All authors have read and approved the content of the manuscript.

Submitted: 22 May 2020

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

# Supplemental material

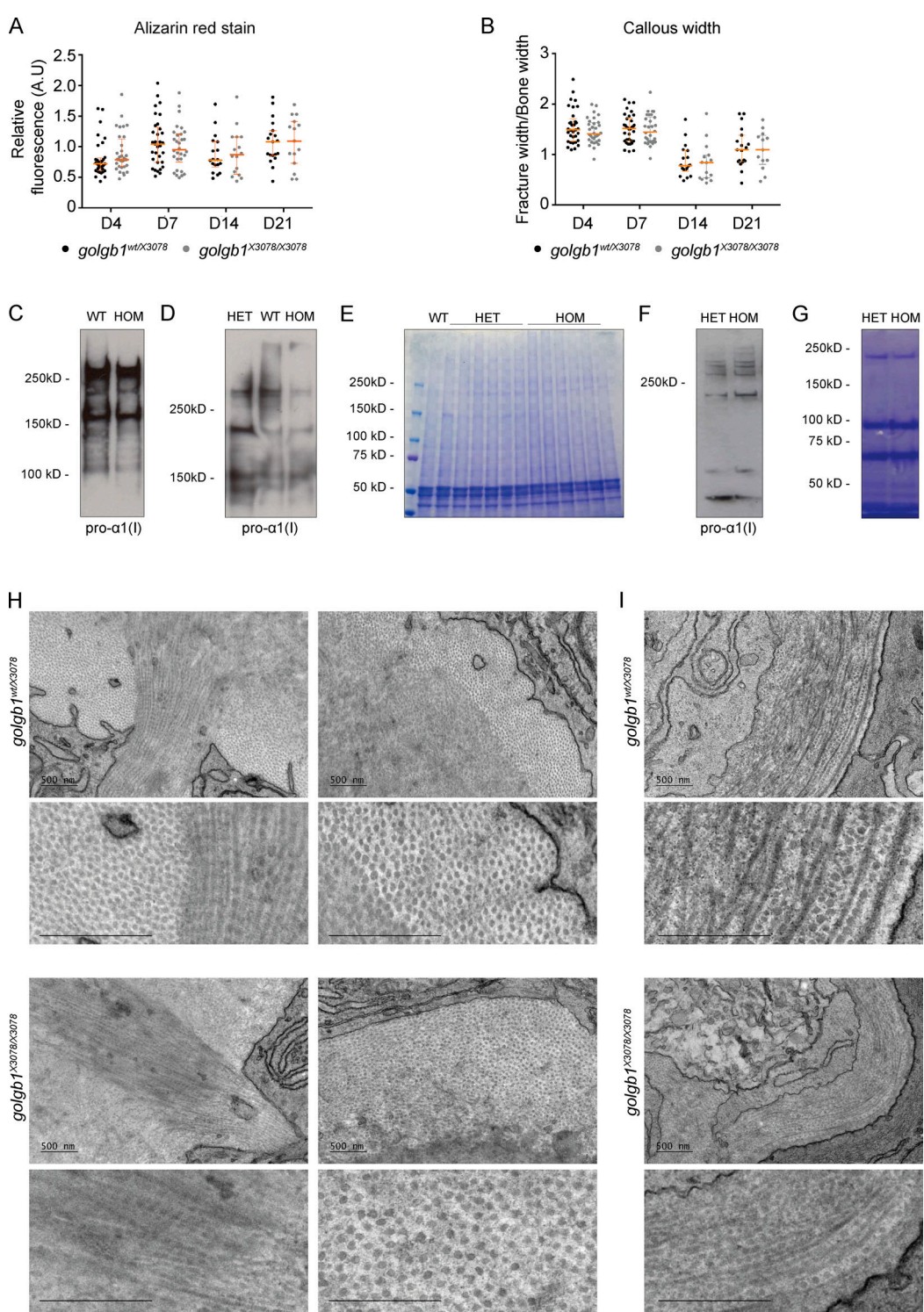

**Figure S1. Fracture analysis in WT and mutant zebrafish. (A and B)** Quantification of the intensity of ARS (A) and callous width of experimentally induced fractures (B) normalized to adjacent healthy bone at different time points after injury (D = days after injury) in *golgb1*<sup>wt/X3078</sup> and *golgb1*<sup>X3078/X3078</sup> fish. Dots represent individual fractures. At time points 4 and 7 dpf, 11 fish per line were quantified. At time points 14 and 21 dpf, *n* = 6 HETs and *n* = 4 HOM fish. All from a single experiment. Horizontal and vertical orange lines depict median and interquartile range, respectively. No significant differences based on P values calculated with a Mann-Whitney *U* test comparing the mean value per fish (three fractures per fish). **(C)** Immunoblots of caudal fin lysates collected from 13-mo-old *golgb1*<sup>+/−</sup>;*col1a1a:GFP+* and *golgb1*<sup>−/−</sup>;*col1a1a:GFP+* fish blotted for pro-α1(I). **(D and E)** Immunoblots with a pro-α1(I) antibody (D) and a Coomassie-stained gel of lysates (E) taken from the regenerated fin of the fish 5 d after initial amputation. **(D)** Each lane shows one representative individual. **(E)** Each lane contains sample from a different individual. **(F and G)** An immunoblot probed with pro-α1(I) (F) and a Coomassie-stained gel of lysates (G) taken from *golgb1*<sup>+/−</sup>; *col1a1a:GFP+* and *golgb1*<sup>−/−</sup>;*col1a1a:GFP+* larvae at 5 dpf. Each sample consists of 20 pooled larvae. **(H and I)** TEM images of collagen fibrils taken at the outer edge of the hemiray bone (H) and basement membrane in the skin (I) of regenerated caudal fins fixed 7 dpi. Fish were 8 mo old at the point of assay. Two females and two males were assayed. All TEM scale bars = 500 nm. A.U, arbitrary units.

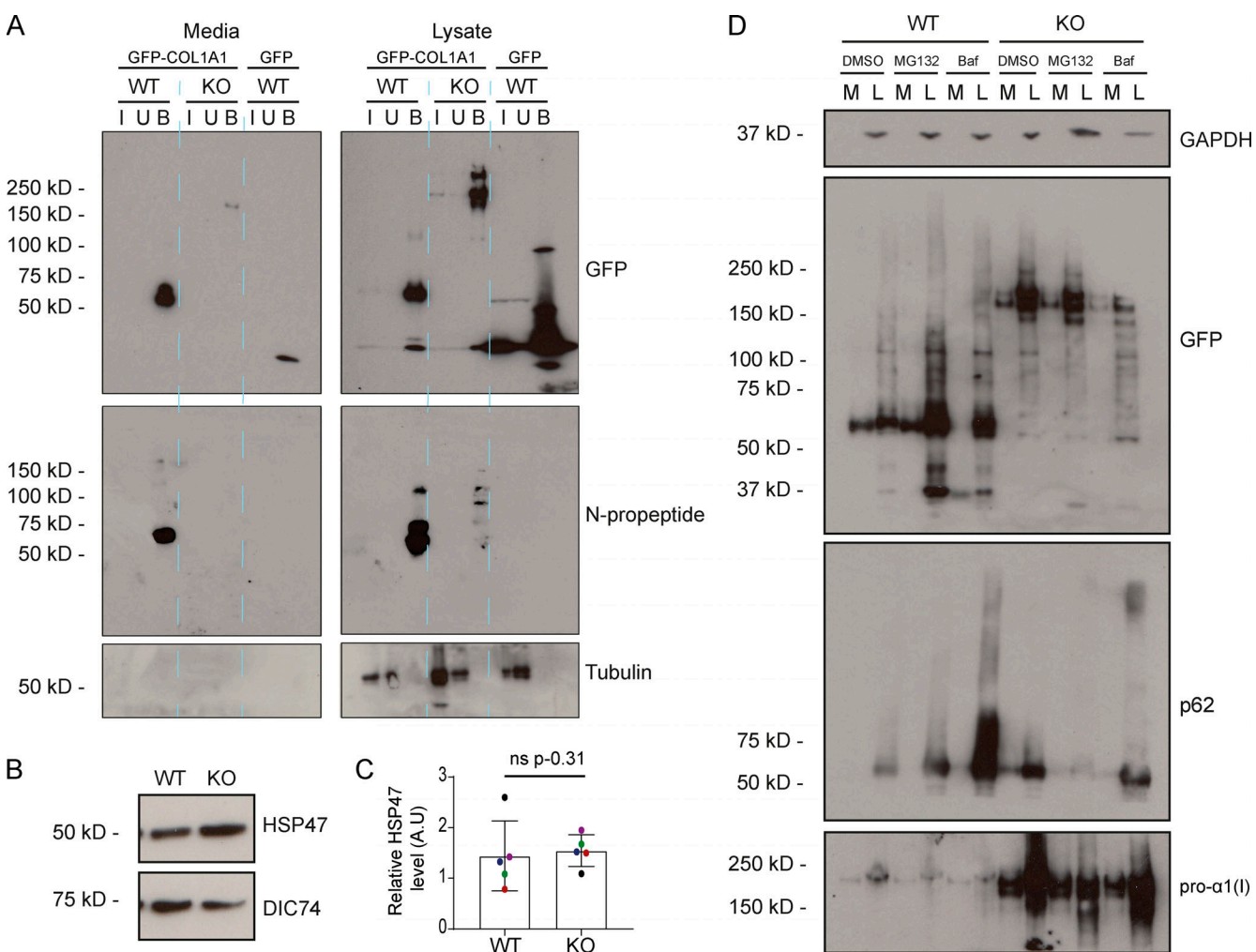

Figure S2. **Procollagen processing controls. (A)** Immunoblot of a GFP trap of media and lysate fractions of WT and giantin KO RPE1 cell cultures. Cells were either expressing GFP-COL1A1 or GFP alone as indicated. Blots show the input (I), unbound (U), and bound (B) fractions of the IP immunoblotted for GFP, pro-α1(I) N-propeptide (LF39 antibody), and tubulin (housekeeping). **(B)** Immunoblot of HSP47 and DIC74 (housekeeping) in WT and giantin KO RPE1 cell lysates. **(C)** Densitometry of the semiquantitative enhanced chemiluminescent immunoblots represented in A. HSP47 levels are normalized to DIC74. Each dot represents an independent biological replicate, and replicates are color coded between cell lines. Bars show median and interquartile range ($n$ = 5 biological replicates). P value calculated with a Mann-Whitney $U$ test. **(D)** Immunoblots of media (M) and lysate (L) fractions of WT and giantin KO RPE1 cell cultures stably expressing GFP-COL1A1 following treatment with DMSO (vehicle control), MG132, or bafilomycin (Baf). Blots are probed for GAPDH (housekeeping), GFP (GFP-COL1A1), p62 (positive control for bafilomycin), and pro-α1(I) (COL1A1) as indicated. A.U, arbitrary units.

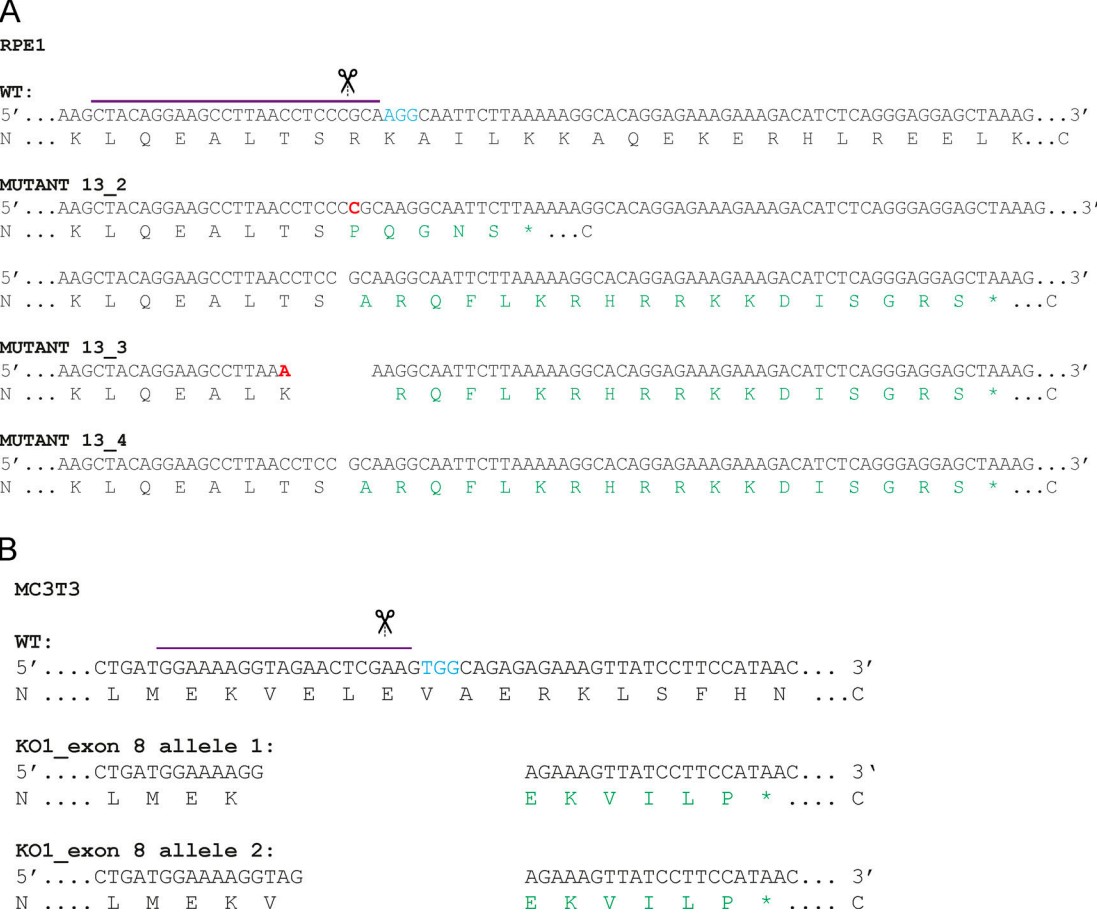

**A**

**RPE1**

**WT:**
```
5'...AAGCTACAGGAAGCCTTAACCTCCCGCAAGGCAATTCTTAAAAAGGCACAGGAGAAAGAAAGACATCTCAGGGAGGAGCTAAAG...3'
N ... K  L  Q  E  A  L  T  S  R  K  A  I  L  K  K  A  Q  E  K  E  R  H  L  R  E  E  L  K...C
```

**MUTANT 13_2**
```
5'...AAGCTACAGGAAGCCTTAACCTCCCCGCAAGGCAATTCTTAAAAAGGCACAGGAGAAAGAAAGACATCTCAGGGAGGAGCTAAAG...3'
N ... K  L  Q  E  A  L  T  S  P  Q  G  N  S  * ...C
```

```
5'...AAGCTACAGGAAGCCTTAACCTCC GCAAGGCAATTCTTAAAAAGGCACAGGAGAAAGAAAGACATCTCAGGGAGGAGCTAAAG...3'
N ... K  L  Q  E  A  L  T  S  A  R  Q  F  L  K  R  H  R  R  K  K  D  I  S  G  R  S  * ...C
```

**MUTANT 13_3**
```
5'...AAGCTACAGGAAGCCTTAAA        AAGGCAATTCTTAAAAAGGCACAGGAGAAAGAAAGACATCTCAGGGAGGAGCTAAAG...3'
N ... K  L  Q  E  A  L  K        R  Q  F  L  K  R  H  R  R  K  K  D  I  S  G  R  S  * ...C
```

**MUTANT 13_4**
```
5'...AAGCTACAGGAAGCCTTAACCTCC GCAAGGCAATTCTTAAAAAGGCACAGGAGAAAGAAAGACATCTCAGGGAGGAGCTAAAG...3'
N ... K  L  Q  E  A  L  T  S  A  R  Q  F  L  K  R  H  R  R  K  K  D  I  S  G  R  S  * ...C
```

**B**

**MC3T3**

**WT:**
```
5'....CTGATGGAAAAGGTAGAACTCGAAGTGGCAGAGAGAAAGTTATCCTTCCATAAC... 3'
N .... L  M  E  K  V  E  L  E  V  A  E  R  K  L  S  F  H  N ...C
```

**KO1_exon 8 allele 1:**
```
5'....CTGATGGAAAAGG              AGAAAGTTATCCTTCCATAAC... 3'
N .... L  M  E  K              E  K  V  I  L  P  * .... C
```

**KO1_exon 8 allele 2:**
```
5'....CTGATGGAAAAGGTAG           AGAAAGTTATCCTTCCATAAC... 3'
N .... L  M  E  K  V           E  K  V  I  L  P  * .... C
```

Figure S3.  **Validating pro-α1(I) processing defect in other giantin KO clones. (A and B)** Genomic DNA sequence and amino acid translation at CRISPR/Cas9 mutation site in three new giantin KO clones targeted at exon 13 (A) and two new MC3T3 mutants targeted at exon 8 (B). On the WT sequences, the gRNA sequence is indicated with a purple line, the cut site is indicated by scissors, and the PAM site is in blue text. In mutant sequences, red text indicates inserted base pairs in genomic sequence, and green text shows altered amino acids. Asterisks denote premature stop codons.

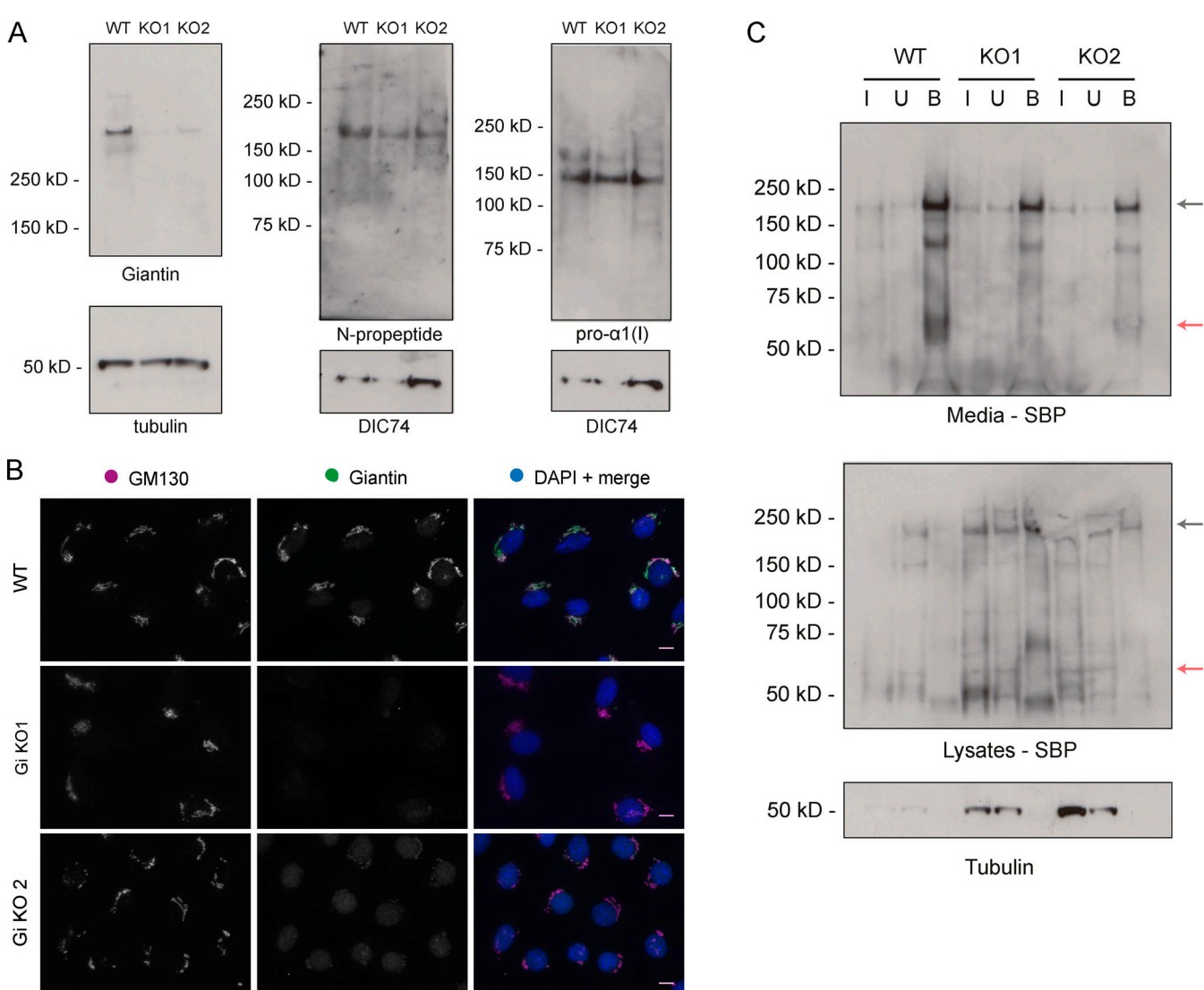

Figure S4. **Procollagen processing in giantin KO MC3T3 cells. (A)** Immunoblots of cell lysates taken from WT and giantin KO MC3T3 cells. Antibodies used are as indicated. **(B)** Single-plane widefield images of WT and giantin KO MC3T3 cells immunolabeled for giantin (green) and the cis-Golgi marker GM130 (magenta). The nucleus is stained in DAPI. Scale bar = 10 μm. **(C)** Immunoblots of a GFP trap of WT and giantin KO MC3T3 cell cultures transiently expressing GFP-COL1A1. Media and lysates were assayed after 24 h of ascorbate treatment, which was conducted 24 h after transfection. The input (I), unbound (U), and bound (B) fraction of each pull-down is shown probed with either SBP or tubulin antibodies as indicated underneath each blot. Black arrows indicate full-length procollagen, and red arrows the N-propeptide.

Video 1. **Representative video of a RUSH assay in a WT RPE1 cell stably expressing GFP-COL1A1 (green) and transiently transfected with mCh-ST (magenta, Golgi label) and an ER RUSH hook.** At time 0, biotin and ascorbic acid were added to the medium to release the collagen from the hook and promote procollagen folding, respectively, to trigger anterograde trafficking. Video is cropped to the relevant time frame; time is indicated in top left corner as min:s. Frames were captured as single-frame images every 20 s. Video frame rate: 12 frames/s. Scale bar = 10 μm.

Video 2. **Representative video of a RUSH assay in a giantin KO RPE1 cell stably expressing GFP-COL1A1 (green) and transiently transfected with mCh-ST (magenta, Golgi label) and an ER RUSH hook.** At time 0, biotin and ascorbic acid were added to the medium to release the collagen from the hook and promote procollagen folding, respectively, to trigger anterograde trafficking. Video is cropped to the relevant time frame; time is indicated in top left corner as min:s. Frames were captured as single-frame images every 20 s. Video frame rate: 12 frames/s. Scale bar = 10 μm.

