## [Peer Review File · The Journal of Cell Biology]

Giantin is required for intracellular N-terminal processing of type I procollagen

Nicola Stevenson, Dylan Bergen, Yinhui Lu, M. Esther Prada-Sanchez, Karl Kadler, Chrissy Hammond, and David Stephens

Corresponding Author(s): David Stephens, University of Bristol and Nicola Stevenson, University of Bristol

Review Timeline:

Submission Date:	2020-05-22
Editorial Decision:	2020-07-06
Revision Received:	2021-01-22
Editorial Decision:	2021-03-04
Revision Received:	2021-03-12

Monitoring Editor: Sean Munro

Scientific Editor: Andrea Marat

Transaction Report:

DOI: <https://doi.org/10.1083/jcb.202005166>

July 6, 2020

Re: JCB manuscript #202005166

Prof. David J Stephens
University of Bristol
Cell Biology Laboratories School of Biochemistry Biomedical Sciences Building
Bristol BS8 1TD
United Kingdom

Dear Prof. Stephens,

Thank you for submitting your manuscript entitled "Giantin is required for intracellular N-terminal processing of type I procollagen". The manuscript was assessed by expert reviewers, whose comments are appended to this letter. We invite you to submit a revision if you can address the reviewers' key concerns, as outlined here.

As you will see, the reviewers agree on the potential importance of the identification of a role for Giantin in procollagen processing in the Golgi. However, given the lack of insight into the mechanism by which giantin contributes to this processing, I agree with the reviewers that more data is required to substantiate your claims. Therefore, in revising the paper you must address their comments with new experimental data where requested.

GENERAL GUIDELINES:

Text limits: Character count for an Article is < 40,000, not including spaces. Count includes title page, abstract, introduction, results, discussion, acknowledgments, and figure legends. Count does not include materials and methods, references, tables, or supplemental legends.

Figures: Articles may have up to 10 main text figures. Figures must be prepared according to the policies outlined in our Instructions to Authors, under Data Presentation, <http://jcb.rupress.org/site/misc/ifora.xhtml>. All figures in accepted manuscripts will be screened prior to publication.

Supplemental information: There are strict limits on the allowable amount of supplemental data. Articles may have up to 5 supplemental figures. Up to 10 supplemental videos or flash animations are allowed. A summary of all supplemental material should appear at the end of the Materials and methods section.

As you may know, the typical timeframe for revisions is three to four months. However, we at JCB realize that the implementation of social distancing and shelter in place measures that limit spread of COVID-19 also pose challenges to scientific researchers. Lab closures especially are preventing scientists from conducting experiments to further their research. Therefore, JCB has waived the revision time limit. We recommend that you reach out to the editors once your lab has reopened to decide on an appropriate time frame for resubmission. Please note that papers are generally considered through only one revision cycle, so any revised manuscript will likely be either accepted or rejected.

Thank you for this interesting contribution to Journal of Cell Biology. You can contact us at the journal office with any questions, cellbio@rockefeller.edu or call (212) 327-8588.

Sincerely,

Sean Munro, PhD
Monitoring Editor

Andrea L. Marat, PhD
Senior Scientific Editor

Journal of Cell Biology

Reviewer #1 (Comments to the Authors (Required)):

The manuscript by Stevenson et al describes experiments illustrating that giantin is required for intracellular processing of the aminopropeptide of type I procollagen.

These findings are of high interest and in the line of previous studies showing that collagen fibril formation can start early during the secretion process.

The data related to bone fragility in the zebrafish model are convincing, but the experiments demonstrating that this phenotype is linked to defective excision of the aminopropeptide of type I procollagen are less persuasive.

The absence (or reduction) of procollagen processing has not been shown in vivo in bone. I understand that many investigating tools are not available for fish models. However, it should be feasible to extract and semi-purify sufficient amounts of collagen from EDTA-decalcified fish bone. This would allow to perform SDS-PAGE and Coomassie blue staining to semi-quantify type I collagen and determine if there are defects in its processing.

Transmission electron microscopy on giantin-KO fish tissues (bone, skin) would have allowed the authors to see if collagen fibrils have highly irregular contours as in dermatosparactic type of EDS

(or EDSVIc).

Since mouse and rat models of giantin-KO have been produced, it would have been interesting to see if these animals have bone related problems and/or decreased bone mineral density.

It has to be stressed that complete absence of ADAMTS2 (human patients and animal models) results in skin fragility because of accumulation of type I aminoprocollagen hampering the formation of collagen fibrils. In the specific case of bone, there is no marked accumulation of type I aminoprocollagen and no obvious osteoporosis/bone fragility. This shows that the processing of fibrillary procollagens can drastically differ from one tissue to another, illustrating the importance of the experimental models. In this context, it is not clear why (except because the model was already available) the author used RPE cells for the study of procollagen processing. Indeed, to the best of my knowledge, the main function of these cells in vivo is not collagen secretion, and, as reported in the manuscript, they do not produce Adamts2 which is much more active and abundant than Adamts3 and 14, at least in mammals. However, the situation could be different in fish where an additional fourth aminoprocollagen peptidase has been described.

The authors have verified their observations made on a first RPE cell line by using other RPE cells, which was of course important. However, it would have been more relevant to use other cell types, preferably of mesenchymal origin (as osteoblasts) which produce large amounts of fibrillary collagen.

The data reported in Fig 2b are not fully convincing for three reasons:

- There is no quantification while the reported differences are not striking
- Collagen fibril formation in culture reflects only very partially fibril formation in vivo
- Many different causes other than aminoprocollagen processing can modify collagen fibril formation (cell confluence, presence of proteoglycans, presence of other components of the extracellular matrix such as fibronectin).

The description of the clinical phenotype of patients with EDS type VII is not accurate. EDSVIIA and VIIB patients (now referred as arthrochalasia) have joint hypermobility, dislocations and skin hyperextensibility (as reported in the manuscript). On the other hand, EDSVIIC patients (now referred as Dermatosparactic type) are characterized by hyper-fragile skin and varying fragility of other soft tissues (blood vessels, bladder, ...). The shape of collagen fibrils is also markedly different between dermatosparaxis (ribbon-like and hieroglyphic pattern) and arthrochalasia (irregular contour).

Reviewer #2 (Comments to the Authors (Required)):

In this article, authors show a new function of giantin using mutant zebrafish and KO cells. At first, it is shown that giantin mutation causes multiple spontaneous fractures, increased mineralization, and diminished procollagen reporter expression in the caudal fin of zebrafish. At second, it is demonstrated that intracellular N-propeptide processing of pro- $\alpha 1(1)$ is defective in the giantin-deficient cells. From these results, authors concluded the presence of a giantin-dependent pathway for intracellular procollagen processing.

Major comments:

This study for the first time presents important information about the function of giantin for

processing of procollagen. However, I would like to confirm several concerns before complete acceptance.

1. I think that the relationship between the results from zebrafish fin and KO cell experiments is slightly unclear. Authors discuss the feedback mechanism of N-propeptide to procollagen synthesis. However, procollagen synthesis is reduced in the mutant fin but increased in KO cells. Is the phenotype of zebrafish fin such as reduced procollagen expression and hypermineralization unrelated with defective N-terminal processing of type I collagen?

2. In this experiment, defective processing of N-propeptide is shown using the transgene of procollagen. Is there any evidence supporting similar phenomenon occurring in endogenous gene expression? Serum procollagen type I N-terminal propeptide concentration sometime increases in several situations. How are important the intracellular N-terminal processing in vivo? In addition, authors focus on type I collagen. Is the enzyme or mechanism related with processing of N-propeptide specific for type I collagen? This mean whether any other collagens are influenced from giantin mutation by similar mechanism.

Minor comments:

1. In line 260, authors describe "mutant fishes show mineralization defects". However, calcification increases in the mutant fish. In this case, the term of defect is OK?

2. In line 535-537 (the legend of Figure 2), (A) and (B) may be changed to (F) and (G), respectively.

Reviewer #1:

The manuscript by Stevenson et al describes experiments illustrating that giantin is required for intracellular processing of the aminopropeptide of type I procollagen.

These findings are of high interest and in the line of previous studies showing that collagen fibril formation can start early during the secretion process.

We thank the reviewer for these positive comments.

The data related to bone fragility in the zebrafish model are convincing, but the experiments demonstrating that this phenotype is linked to defective excision of the aminopropeptide of type I procollagen are less persuasive. The absence (or reduction) of procollagen processing has not been shown in vivo in bone. I understand that many investigating tools are not available for fish models. However, it should be feasible to extract and semi-purify sufficient amounts of collagen from EDTA-decalcified fish bone. This would allow to perform SDS-PAGE and Coomassie blue staining to semi-quantify type I collagen and determine if there are defects in its processing.

We thank the reviewer for raising this issue. As suggested above, it has been difficult to directly link the processing defects seen in the cells with the in vivo phenotypes, in part because of tools.

At the reviewer's suggestion, we tried performing Coomassie stains of lysates from dissected adult caudal fins (primarily bone and some skin) and lysates from 5 d.p.f larvae. Unfortunately, there were no bands that could be confidently identified as collagen, despite the abundance of bone in the sample. We thus do not feel we can interpret these data usefully.

We did, however, manage to find an antibody against type I collagen which worked for immunoblots. This failed to show a difference in high molecular weight collagen bands between WT/heterozygous and homozygous mutant fish, but this replicates the cell results – mature collagen was still detectable in the cell knockouts and we could only discern a difference by blotting for the propeptide specifically. Unfortunately, we were unable to find an antibody against the N-propeptide for fish samples.

To be thorough, we decided to test immature tissue to see if it contained detectable levels of unprocessed procollagen, the logic being that compensatory extracellular processing may occur during tissue maturation and mask the giantin phenotype. We amputated the caudal fins and then allowed the fin to regenerate for 5 days before amputating again to lyse the regenerated tissue and probe for type I collagen. Again, we could not see a difference between the genotypes as there was an abundance of mature collagen.

As we state in the discussion, we do not think the presence of an intracellular pathway of procollagen processing precludes the presence of an extracellular one, which could compensate by cleaving the N-propeptide during tissue remodelling. Therefore we do not believe that these data disprove our model of giantin function, but agree that so far we have been unable to link the fish skeletal phenotypes with a procollagen processing defect. We acknowledge that this is a limitation of the study and have stated this clearly in the manuscript discussion in lines 636-638.

We have also included this data in the manuscript. See lines 367-379 and Supplemental figure 1.

Transmission electron microscopy on giantin-KO fish tissues (bone, skin) would have allowed the authors to see if collagen fibrils have highly irregular contours as in dermatosparatic type of EDS (or EDS VIIc).

We thank the author for this suggestion. To look at this we have now performed TEM on the skin and bones of the caudal fin of our fish lines (with our colleagues Kadler and Lu in Manchester who are now included as co-authors). Overall, there was no striking difference between the two genotypes, but there was a lot of variability between samples that made it difficult to compare fibril density, diameter, etc. We did get sufficiently clean cross sections, however, to determine that the contours are largely regular in both the bone and the basement membrane of the skin. Collagen periodicity also appeared normal. Unfortunately, these data still do not help us explain the fractures and mineralisation defects in the mutants so this will require further investigation. We have included these EM data in the manuscript in supplemental figure 1 and lines 381-388. Discussion of this in relation to EDS can be found in lines 603-607.

Structurally, fish type I collagen is slightly different to human as there are 3 alpha chains. Furthermore, to our knowledge the procollagen cleavage site should be intact in the giantin mutants and we predict that at least some processing enzymes are still present and active in the mutant. Thus, loss of giantin may present differently to complete loss of, for example ADAMTS2, because it is acting more indirectly on the efficiency of the processing machinery.

Since mouse and rat models of giantin-KO have been produced, it would have been interesting to see if these animals have bone related problems and/or decreased bone mineral density.

We agree that this would have been interesting to investigate, however we have not been able to access these models and such investigations are limited by the fact that neither the rat nor mouse models survive post-natally. Rat embryos do show quite extreme skeletal and mineralisation defects, whereas the mice have cleft palate. Some limited investigation into bone development and endochondral ossification has been performed on rat mutant pups and at E20.5 there are some small signs of advanced mineralisation in the calvaria specifically (Katayama et al. Bone 49 (2011) 1027–1036), however little else of direct relevance is known.

It has to be stressed that complete absence of ADAMTS2 (human patients and animal models) results in skin fragility because of accumulation of type I aminoprocollagen hampering the formation of collagen fibrils. In the specific case of bone, there is no marked accumulation of type I aminoprocollagen and no obvious osteoporosis/bone fragility. This shows that the processing of fibrillary procollagens can drastically differ from one tissue to another, illustrating the importance of the experimental models. In this context, it is not clear why (except because the model was already available) the author used RPE cells for the study of procollagen processing. Indeed, to the best of my knowledge, the main function of these cells in vivo is not collagen secretion, and, as reported in the manuscript, they do not produce Adamts2 which is much more active and abundant than Adamts3 and 14, at least in mammals. However, the situation could be different in fish where an additional fourth aminoprocollagen peptidase has been described.

The authors have verified their observations made on a first RPE cell line by using other RPE cells, which was of course important. However, it would have been more relevant to use other cell types, preferably of mesenchymal origin (as osteoblasts) which produce large amounts of fibrillary collagen.

We accept that RPE1 cells are not necessarily the best choice to study type I collagen but our initial work related to the wider ECM. As a non-transformed but immortalised, diploid, human cell line these cells are well suited to our CRISPR workflow and in our hands the RPE1 cells make a more reliable matrix than the immortalised fibroblast lines that we have tried (some evidence of this can be seen in our previous work, McCaughey et al 2019). Since bones (the most affected tissue in giantin KO animal models) are less affected by the loss of ADAMTS2 compared to other tissues (Giunta et al J Bone Joint Surg Am. 1999 Feb;81(2):225-38.) the absence of ADAMTS2 in the RPE1 cells was not a major concern to us as it seemed likely there was another mechanism at play. Nonetheless, we have addressed this point by creating two giantin KO MC3T3 cell lines. MC3T3 cells are an osteoblast precursor cell line and thus we hope the reviewer will consider them more relevant to the phenotypes described. They are also of mouse origin and thus allow us to test another system.

Interestingly, we found that the giantin KO MC3T3 cells show an N-propeptide processing defect as expected, however it manifested slightly differently. In agreement with the RPE1 cells, we found that when we expressed our pro-SBP-GFP-COL1A1 construct in the MC3T3 lines, the GFP-tagged N-propeptide was only detectable in the media of WT cells and not KO cells. Unlike the RPE1 however, some cleaved N-propeptide is detectable in the lysates of the KO cells. Whether this is in the secretory or degradative pathways has not been determined but since this product is not secreted it could be the latter. It is also important to note that in these cell lines we did not achieve a complete loss of protein. and that we have expressed a human protein in a mouse line which may be handled or recognised slightly differently by the mouse machinery and enzymes. It is also likely that different cell types may organise their secretory and processing machinery differently as the authors point out.

These data are described in lines 514-534 and Supplemental Figure 4 and are discussed in lines 576-585.

The data reported in Fig 2b are not fully convincing for three reasons:

- There is no quantification while the reported differences are not striking
- Collagen fibril formation in culture reflects only very partially fibril formation in vivo
- Many different causes other than aminoprocollagen processing can modify collagen fibril formation (cell confluence, presence of proteoglycans, presence of other components of the extracellular matrix such as fibronectin).

We think that the reviewer is referring to figure 2F and so have addressed this comment with respect to those data.

We agree that analysis of fibril formation in 2D cell cultures is limited in use and we did not include any quantification as it was too challenging to robustly isolate fibres for automated, unbiased analysis. Now that we have high quality EM data from the fish, and therefore physiologically relevant fibrils, we have removed these data from the figure as they could not be improved upon. We have left in the immunofluorescence of collagen to illustrate that it is assembling into fibrils in the extracellular matrix.

The description of the clinical phenotype of patients with EDS type VII is not accurate. EDSVIIA and VIIB patients (now referred as arthrochalasia) have joint hypermobility, dislocations and skin hyperextensibility (as reported in the manuscript). On the other hand, EDSVIIC patients (now referred as dermatosparatic type) are characterized by hyper-fragile skin and varying fragility of other soft tissues (blood vessels, bladder, ...). The shape of collagen fibrils is also markedly different between dermatosparaxis (ribbon-like and hieroglyphic pattern) and arthrochalasia (irregular contour).

We apologise for over-simplifying these conditions, which is not our area of expertise, and thank the reviewer for correcting us to ensure our descriptions are accurate. We have amended the text accordingly and hopefully to the reviewer's satisfaction.

Reviewer #2

In this article, authors show a new function of giantin using mutant zebrafish and KO cells. At first, it is shown that giantin mutation causes multiple spontaneous fractures, increased mineralization, and diminished procollagen reporter expression in the caudal fin of zebrafish. At second, it is demonstrated that intracellular N-propeptide processing of pro- α 1(1) is defective in the giantin-deficient cells. From these results, authors concluded the presence of a giantin-dependent pathway for intracellular procollagen processing.

Major comments:

This study for the first time presents important information about the function of giantin for processing of procollagen. However, I would like to confirm several concerns before complete acceptance.

1. I think that the relationship between the results from zebrafish fin and KO cell experiments is slightly unclear. Authors discuss the feedback mechanism of N-propeptide to procollagen synthesis. However, procollagen synthesis is reduced in the mutant fin but increased in KO cells. Is the phenotype of zebrafish fin such as reduced procollagen expression and hypermineralization unrelated with defective N-terminal processing of type I collagen?

As the reviewer points out, our *in vivo* data here show that COL1A1A promoter activity, and therefore transcription, is reduced in the mutant fish. In our previous paper, Stevenson et al 2017, we performed RNAseq analysis of the giantin KO RPE1 and found that mRNA levels were also reduced in the KO cells. So with respect to mRNA, the two systems do agree.

The increased procollagen synthesis in cells shown in the current study and highlighted by the reviewer is of protein levels rather than mRNA. So, although transcription is reduced, protein is increased. This could be explained by an increase in protein translation or reduction in degradation. Previous studies imply that the N-propeptide could feedback directly on translation and thus can influence protein levels without affecting mRNA. We therefore propose that in the absence of N-propeptide in the knockout cells, there is no negative feedback on procollagen translation and thus more protein gets made. The presence of more mature collagen/fibrils may then feedback to reduce mRNA expression to compensate.

We have tried to make it clearer in the text when we are talking about mRNA or protein – line 586

This is of course speculation as we have been unable to test this model. As the reviewer points out, we cannot currently directly link the processing defect with the phenotypes observed in fish and so there may be alternative explanations, such as changes in other matrix molecules, glycosylation etc which may be affected by disruption to Golgi function.

To address the reviewers concerns we have stated more clearly that this is speculation in line 592.

2. In this experiment, defective processing of N-propeptide is shown using the transgene of procollagen. Is there any evidence supporting similar phenomenon occurring in endogenous gene expression?

We have spent a lot of time trying to validate our model by probing endogenous protein but have so far been unsuccessful as our antibody is too insensitive to detect the low levels of endogenous N-propeptide likely present. In the case of RPE1 cells, our N-propeptide was unable to detect any endogenous procollagen or N-propeptide, however in the MC3T3 cells we could detect the full length procollagen (likely in high abundance) but still not the N-propeptide alone (low abundance). We have therefore been unable to address this but have included the endogenous MC3T3 blot in Supplemental Figure 4.

Serum procollagen type I N-terminal propeptide concentration sometime increases in several situations. How are important the intracellular N-terminal processing *in vivo*?

The importance of the processing event itself is best demonstrated in patients with mutations in COL1A1 affecting the cleavage site or mutations in the processing enzymes themselves, primarily ADAMTS2. These cause some subtypes of Ehlers Danlos syndrome characterised by hyperelastic skin and hypermobile joints etc. This is largely because of the effect its retention has on collagen fibril structure in tissues. It is unknown how important intracellular vs extracellular cleavage is as intracellular processing has not been studied. Our fish may provide information on this but so far we have been unable to link processing to our phenotypes. Also very little is known about whether circulating type I collagen N-propeptide (PINP) has any function in

its own right, and so the importance of its presence in serum remains elusive. Levels of serum PINP change with bone turnover but this may just be a by-product of collagen secretion and degradation.

In addition, authors focus on type I collagen. Is the enzyme or mechanism related with processing of N-propeptide specific for type I collagen? This mean whether any other collagens are influenced from giantin mutation by similar mechanism.

Given that bone is the most affected tissue in the animal models type I collagen is likely to be the primary, or most sensitive, client of this pathway. We would love to know whether other collagens or matrix molecules are affected but so far our attempts to develop tools to investigate this have not been successful. Collagen type II and III are processed by similar enzymes in vivo and so these are potential targets of study. We will continue to try to investigate this but consider it outside the scope of the current study.

Minor comments:

1. In line 260, authors describe "mutant fishes show mineralization defects". However, calcification increases in the mutant fish. In this case, the term of defect is OK?

We have now clarified this by using the term 'abnormal mineralisation patterns'

2. In line 535-537 (the legend of Figure 2), (A) and (B) may be changed to (F) and (G), respectively.

The Figure 2 legend has been changed.

Reviewer #3

In this manuscript from Stevenson et al. report on the involvement of giantin in secretion and processing of type I collagen. This sheds new light on the role of Giantin in bone development and help to understand why lack of proper giantin function leads to skeletal and craniofacial defects.

This is a well-done and interesting study. Experiments are carefully done and exploited. Altogether, I think that it would be acceptable for publication in a short format in the JCB. I however have several comments, a few of them being quite important to wrap up the analysis.

We thank the reviewer for these positive comments. Given the amount of extra data we have been asked to provide, we have again submitted the manuscript as a full article but should a short report still be considered more appropriate we are happy to discuss this. The 5 figure, 20000 character limit would be a challenge to meet.

1- My main comment concerns the analysis of COL1A1 cleavage which central in the discussion. The authors favor a model by which N-terminal cleavage would occur inside the Golgi and that this particular step would be dependent on giantin activity. I do not think that the authors convincingly show this point. The way they realize "cellular extract" does not ensure that no extracellular proteins/peptides are extracted. I think that they have in hands all the tool to study this.

They indeed implemented pulsed trafficking of COL1A1 and can thus study when cleavage occurs.

We have looked into this. Unfortunately, in the RUSH system the cells do not synchronise sufficiently to achieve the fine temporal resolution required for biochemistry experiments, and the N-propeptide antibodies do not work for us on fixed cells. However, we believe we have addressed this issue with the alternative experiments suggested by the reviewer below and thank them for their help in strengthening the conclusions of our study.

Using 20°C temperature block (or using monensin) they may even confirm that Golgi exit is unnecessary to ensure N-terminal processing.

As suggested, we performed secretion assays on the WT RPE expressing GFP-COL1a1 at 20C. First, we confirmed that this did indeed block collagen Golgi exit by imaging the GFP tag. Then we devised a protocol using ascorbate and cycloheximide to flush through any pre-existing collagen that may have been processed before the temperature block to be sure we were looking at processing at 20C only. Then we performed secretion assays of these cells at 37C and 20C to look for processing and confirmed that the N-propeptide is cleaved at 20C even though no collagen or N-propeptide leaves the Golgi.

To even further demonstrate that the N-propeptide we are detecting is intracellular and not stuck on the cell surface or in cell-surface associated matrix, we also did a trypsin surface digest of the cells. We found the N-propeptide was protected from trypsin cleavage, even in the presence of digitonin, as was HSP47 but not the EGFR. Thus, we believe it resides in the digitonin resistant membranes of the secretory pathway and not at the cell surface or in endosome.

Altogether we hope this convinces the reviewer that we are looking at intracellular processing. These data can now be found in Figure 5 of the manuscript and are described in lines 465-494.

If I am not mistaken, the analysis by WB of the RUSH COL1A1 (GFP-COL1A1) upon biotin addition has not been analyzed in McCaughey et al 2019. Such an analysis would allow to understand better where the cleavage of the propeptides occur and will refine the analysis of the effects of GOLGB1 KO.

This analysis would be ideal but is precluded by the large pool of ER-localized reporter which reflects that of endogenous collagen in these cells. We consider that our additional experiments at 20°C now address this to some extent.

The authors indicate in the discussion «We also found that free N389 propeptide is clearly detected in the cell layer of WT cells, implying it is cleaved inside the cell.» However, the N-propeptide may be cleaved at the cell surface (via plasma membrane enzymes for example). The cell layer might indeed contain deposited ECM (the authors analyzed media vs. cell layer).

We hope the above experiments have satisfied the author in this respect. Please refer to the new data in Figure 5.

2- In Asante et al. 2013 JCS, the same group reported the effect of giantin depletion (siRNA) on ciliogenesis by affecting the intermediate chain of dynein-2 (its localization, expression level) was not analyzed. In this context, maybe another loading control than DIC74 (dynein intermediate chain) should be used for quantification of protein levels.

DIC74 is part of the cytoplasmic dynein-1 motor which is independent of the ciliary dynein-2 motor and is not affected by giantin KO. When BCA assays are used to normalise protein loading there is no difference between DIC74 levels in WT and KO cells or between DIC74 and other housekeeping proteins probed simultaneously so we are confident it is not affected. The use of DIC74 as a house-keeping protein was often used preferentially as it is of a higher molecular weight than many other commonly used housekeeping proteins which suits the well-resolved gels used to separate collagen. It is also suitably distanced from the N-propeptide band to allow us to probe both collagen and housekeeping without the need for stripping buffers which may add more variability. In many assays we did use alternative housekeeping proteins on the same assays and they gave the same results as DIC74. We hope that this convinces the reviewer that it is an adequate control.

3- Why only HET and HOM were analyzed and not golgb1 wt/wt in some experiments? Please comment.

As you can see in figure 1 and also in our previous papers (Bergen et al) the heterozygotes have no discernible phenotypes. To comply with the ethical considerations encompassed by the philosophy of the NC3Rs (reduce, refine, replace) we chose to make use of all the animals (siblings) that were available to us from as few crosses as possible.

4- In Figure 2, please clearly indicate in the legend that KO cells are RPE-1 cells GOLGB1 KO cells.

The cells are now identified as RPE1 cells in the figure title.

5- Fig. 2B corresponds to the quantification of Fig. 2A. However, on the panel B the 2 bars are indicated as « HET » and « HOM » while it should be WT and KO. Please confirm that the bars displayed actually correspond to WT and KO and please modify the panel. Same comment for fig 2D corresponding to quantification of 2C

We thank the reviewer for spotting this error. They should indeed be WT and KO and this has been amended.

6- Fig 2 F and G: quantification of the effects would be helpful

We agree this would be useful however after a few attempts we found it was not possible to isolate fibres for automated, unbiased analysis from these data sets. In light of the fact we now have high quality EM data from the fish, and therefore more physiologically relevant fibrils than those formed in 2D culture, we have

removed the cell SEM data from the manuscript. We have left in the immunofluorescence of collagen as representative images to illustrate that it is assembling into fibrils in the extracellular matrix but have now stated that we couldn't quantify it. Please see lines 403-407 and Figure 2 for these amendments.

7- Fig4C: if possible, please provide another picture of SBP immunoblot. The band corresponding to the unprocessed form (dominant in KO cells) is not clearly visible.

Another blot has been provided.

8- Fig4: Please provide more details about the plasmid coding for GFP-COL1A1 used in this figure. Is this the SBP-GFP-COL1A1? If yes, indicate if an ER hook is co-expressed and also mention if biotin is added (if yes, for how long).

There is no hook in these cells and therefore there was no need for biotin. This has now been explicitly stated in the figure legend. The methods also detail all the treatments for the secretion assays, which do not include transfections of a hook or biotin.

Please also comment on the localization of the intracellular GFP signal of GFP-COL1A1 in regard of the experimental conditions. This is also needed in Figure 5.

The GFP signal is predominantly seen in the ER in the experiments shown in figure 4 and 5 as is typical for endogenous collagen in these cells (see McCaughey 2019). In the absence of a hook, the GFP-COL1A1 construct should behave the same way as endogenous procollagen and the main pool of procollagen is the ER.

9- Fig4E-G: please provide a loading control or clearly state that the same membrane is used for all panels from B to G.

We have added in loading controls for the majority of the blots – in each case the same membrane was re-probed for the housekeeping gene. In the case of the N-propeptide we really struggled to achieve these blots because of the quality of the antibodies - we wanted to collect all processing forms, and so the whole membrane was blotted and the high antibody concentrations used tended to mean that subsequent re-probing of the membranes with housekeeping genes was messy and unsuccessful. Loading therefore had to be checked by Ponceau S for which we do not have images. We understand the reviewers concerns about this, however we have not sought to quantify these data and we have full confidence our results are correct with respect to the presence or absence of peptide as they were highly reproducible (you can see they repeat in supplemental figure 2 and figure 6). We therefore hope these limitations can be accepted now that the figure is amended to include housekeeping proteins for the other individual blots.

10- As a comment/warning: Unless this is a novel antibody, I do not think that the anti-giantin used here and obtained from the lab of M. Lowe has been raised against the C-ter of giantin. Could the authors double-check? It is important when using it to ensure that no "small" giantin is expressed upon genome editing.

We apologise for this error. This antibody was raised against the whole giantin protein as described in Lowe et al 2004 JCS: “*Polyclonal antibodies raised against full-length giantin were from Prof. Manfred Renz (Institute of Immunology and Molecular Genetics, Karlsruhe, Germany)*” . This has been amended in the methods section of our manuscript (line 202) and the results section (line 502), and we now also credit Prof Renz for the antibody as we should have in the first place.

The premature stop codons present in our mutants should only permit expression of truncated giantin peptides which lack the C-terminal transmembrane domain required for function, if indeed any protein is translated at all. However, in the absence of a C-terminal specific antibody, we cannot entirely rule out the expression of alternative N-terminally truncated gene products transcribed from alternative start sites. Given that we see a phenotype we believe giantin function is disrupted in these cells.

March 4, 2021

RE: JCB Manuscript #202005166R

Prof. David J Stephens
University of Bristol
Cell Biology Laboratories School of Biochemistry Biomedical Sciences Building
Bristol BS8 1TD
United Kingdom

Dear Prof. Stephens:

Thank you for submitting your revised manuscript entitled "Giantin is required for intracellular N-terminal processing of type I procollagen". The reviewers all now support publication so we would be happy to publish your paper in JCB pending final revisions necessary to meet our formatting guidelines (see details below). In your final revision, please be sure to correct all typos and add a higher magnification EM image(s). While we do not require you to redo experiments with different extraction conditions, you should comment in the text on the absence of collagen bands in the gel as noted by reviewer #1.

A. MANUSCRIPT ORGANIZATION AND FORMATTING:

Full guidelines are available on our Instructions for Authors page, <https://jcb.rupress.org/submission-guidelines#revised>. **Submission of a paper that does not conform to JCB guidelines will delay the acceptance of your manuscript.**

1) Text limits: Character count for Articles is < 40,000, not including spaces. Count includes title page, abstract, introduction, results, discussion, acknowledgments, and figure legends. Count does not include materials and methods, references, tables, or supplemental legends.

2) Figures limits: Articles may have up to 10 main text figures.

3) Figure formatting: Scale bars must be present on all microscopy images, including inset magnifications. Molecular weight or nucleic acid size markers must be included on all gel electrophoresis.

4) Statistical analysis: Error bars on graphic representations of numerical data must be clearly described in the figure legend. The number of independent data points (n) represented in a graph must be indicated in the legend. Statistical methods should be explained in full in the materials and methods. For figures presenting pooled data the statistical measure should be defined in the figure legends. Please also be sure to indicate the statistical tests used in each of your experiments (either in the figure legend itself or in a separate methods section) as well as the parameters of the test (for example, if you ran a t-test, please indicate if it was one- or two-sided, etc.). Also, if you used parametric tests, please indicate if the data distribution was tested for normality (and if so,

how). If not, you must state something to the effect that "Data distribution was assumed to be normal but this was not formally tested."

5) Abstract and title: The abstract should be no longer than 160 words and should communicate the significance of the paper for a general audience. The title should be less than 100 characters including spaces. Make the title concise but accessible to a general readership.

We suggest a slightly simplified version of your title:

Giantin is required for intracellular processing of type I procollagen

6) Materials and methods: Should be comprehensive and not simply reference a previous publication for details on how an experiment was performed.

7) Please be sure to provide the sequences for all of your primers/oligos and RNAi constructs in the materials and methods. You must also indicate in the methods the source, species, and catalog numbers (where appropriate) for all of your antibodies. Please also indicate the acquisition and quantification methods for immunoblotting/western blots.

8) Microscope image acquisition: The following information must be provided about the acquisition and processing of images:

a. Make and model of microscope

b. Type, magnification, and numerical aperture of the objective lenses

c. Temperature

d. Imaging medium

e. Fluorochromes

f. Camera make and model

g. Acquisition software

h. Any software used for image processing subsequent to data acquisition. Please include details and types of operations involved (e.g., type of deconvolution, 3D reconstitutions, surface or volume rendering, gamma adjustments, etc.).

10) Supplemental materials: There are strict limits on the allowable amount of supplemental data. Articles may have up to 5 supplemental display items (figures and tables). Please also note that tables, like figures, should be provided as individual, editable files. A summary of all supplemental material should appear at the end of the Materials and methods section.

13) ORCID IDs: ORCID IDs are unique identifiers allowing researchers to create a record of their various scholarly contributions in a single place. At resubmission of your final files, please consider providing an ORCID ID for as many contributing authors as possible.

B. FINAL FILES:

-- High-resolution figure and video files: See our detailed guidelines for preparing your production-ready images, <https://jcb.rupress.org/fig-vid-guidelines>.

Thank you for this interesting contribution, we look forward to publishing your paper in Journal of Cell Biology.

Sincerely,

Sean Munro, PhD
Monitoring Editor

Andrea L. Marat, PhD
Senior Scientific Editor

Reviewer #1 (Comments to the Authors (Required)):

The authors have responded to all my comments and questions. However, I still have some comments.

Regarding collagen visualization on gels:

The absence of collagen bands on their SDS-PAGE after Coomassie blue staining is surprising. It is possibly related to the extraction buffer (RIPA) since usually collagen is extracted from tissues using acidic conditions or heat denaturation in Laemmli sample buffer (Gistelink et al, Scientific Reports). As a result, the amount of recovered collagen is probably quite low and only sufficient for Western blotting analysis.

Observations of collagen fibers by TEM was worth to be added. However, the magnification used for these pictures is too low to allow readers to see them as they are described.

Still a few mistakes in the text:

Line 73: "ADAMTS2, -3 and -4" should be modified in "ADAMTS2, -3 and -14"

Line 369: "col11" should be "col1a1"

Line 602: the authors again inverted "arthrochlasia" and "dermatosparaxis"!!! Fibers have an irregular contour in arthrochlasia (EDSVIIA and B) and are hieroglyphic in dermatosparactic EDS (EDSVIIC)

Line 640: Please "invert arthrochlasia type" and "dermatosparactic-type" in order to be in accordance with the order used on line 641 for the mutations (ADAMTS2 for dermatosparactic, and Collagen for arthrochlasic) .

Line 641: In arthrochlasic EDS, deletion of the domain encompassing the cleavage site by ADAMTS2 can be seen in pro-alpha1(I) as written (EDSVIIA), but also in pro-alpha2(I) (EDSVIIB). Please modify!

Reviewer #2 (Comments to the Authors (Required)):

This is the first report in which giantin is related with intercellular procollagen processing. Electronic microscopic analysis and in vitro experiments were added in revised manuscripts. All experiments were carefully done and analyzed. All of my concerns have been resolved in this revision.

Reviewer #3 (Comments to the Authors (Required)):

In this revised version, the authors addressed all the points I raised before. I would like to congratulate them for this great study.

I am of course supporting publication of this manuscript in the Journal of Cell Biology.